# Dataset Distillation with Infinitely Wide Convolutional Networks

**Timothy Nguyen**[†] [*]  **Roman Novak**[♦]  **Lechao Xiao**[♦]  **Jaehoon Lee**[♦]

DeepMind[†]  Google Research, Brain Team[♦]

timothycnguyen@deepmind.com
{romann, xlc, jaehlee}@google.com

## Abstract

The effectiveness of machine learning algorithms arises from being able to extract useful features from large amounts of data. As model and dataset sizes increase, dataset distillation methods that compress large datasets into significantly smaller yet highly performant ones will become valuable in terms of training efficiency and useful feature extraction. To that end, we apply a novel distributed kernel-based meta-learning framework to achieve state-of-the-art results for dataset distillation using infinitely wide convolutional neural networks. For instance, using only 10 datapoints (0.02% of original dataset), we obtain over 65% test accuracy on CIFAR-10 image classification task, a dramatic improvement over the previous best test accuracy of 40%. Our state-of-the-art results extend across many other settings for MNIST, Fashion-MNIST, CIFAR-10, CIFAR-100, and SVHN. Furthermore, we perform some preliminary analyses of our distilled datasets to shed light on how they differ from naturally occurring data.

## 1 Introduction

Deep learning has become extraordinarily successful in a wide variety of settings through the availability of large datasets [Krizhevsky et al., 2012, Devlin et al., 2018, Brown et al., 2020, Dosovitskiy et al., 2020]. Such large datasets enable a neural network to learn useful representations of the data that are adapted to solving tasks of interest. Unfortunately, it can be prohibitively costly to acquire such large datasets and train a neural network for the requisite amount of time.

One way to mitigate this problem is by constructing smaller datasets that are nevertheless informative. Some direct approaches to this include choosing a representative subset of the dataset (i.e. a coreset) or else performing a low-dimensional projection that reduces the number of features. However, such methods typically introduce a tradeoff between performance and dataset size, since what they produce is a coarse approximation of the full dataset. By contrast, the approach of *dataset distillation* is to synthesize datasets that are *more* informative than their natural counterparts when equalizing for dataset size [Wang et al., 2018, Bohdal et al., 2020, Nguyen et al., 2021, Zhao and Bilen, 2021]. Such resulting datasets will not arise from the distribution of natural images but will nevertheless capture features useful to a neural network, a capability which remains mysterious and is far from being well-understood [Ilyas et al., 2019, Huh et al., 2016, Hermann and Lampinen, 2020].

The applications of such smaller, distilled datasets are diverse. For nonparametric methods that scale poorly with the training dataset (e.g. nearest-neighbors or kernel-ridge regression), having a reduced dataset decreases the associated memory and inference costs. For the training of neural networks, such distilled datasets have found several applications in the literature, including increasing the

---

[*]Work done while at Google Research.

35th Conference on Neural Information Processing Systems (NeurIPS 2021).

effectiveness of replay methods in continual learning [Borsos et al., 2020] and helping to accelerate neural architecture search [Zhao et al., 2021, Zhao and Bilen, 2021].

In this paper, we perform a large-scale extension of the methods of Nguyen et al. [2021] to obtain new state-of-the-art (SOTA) dataset distillation results. Specifically, we apply the algorithms KIP (Kernel Inducing Points) and LS (Label Solve), first developed in Nguyen et al. [2021], to infinitely wide convolutional networks by implementing a novel, distributed meta-learning framework that draws upon hundreds of accelerators per training. The need for such resources is necessitated by the computational costs of using infinitely wide neural networks built out of components occurring in modern image classification models: convolutional and pooling layers (see §B for details). The consequence is that we obtain distilled datasets that are effective for both kernel ridge-regression and neural network training.

Additionally, we initiate a preliminary study of the images and labels which KIP learns. We provide a visual and quantitative analysis of the data learned and find some surprising results concerning their interpretability and their dimensional and spectral properties. Given the efficacy of KIP and LS learned data, we believe a better understanding of them would aid in the understanding of feature learning in neural networks.

To summarize, our contributions are as follows:

1. We achieve SOTA dataset distillation results on a wide variety of datasets (MNIST, Fashion-MNIST, SVHN, CIFAR-10, CIFAR-100) for both kernel ridge-regression and neural network training. In several instances, our results achieve an impressively wide margin over prior art, including over 25% and 37% absolute gain in accuracy on CIFAR-10 and SVHN image classification, respectively, when using only 10 images (Tables 1, 2, A11).

2. We develop a novel, distributed meta-learning framework specifically tailored to the computational burdens of sophisticated neural kernels (§2.1).

3. We highlight and analyze some of the peculiar features of the distilled datasets we obtain, illustrating how they differ from natural data (§4).

4. We open source the distilled datasets, which used thousands of GPU hours, for the research community to further investigate at https://github.com/google-research/google-research/tree/master/kip.

## 2 Setup

**Background on infinitely wide convolutional networks**. Recent literature has established that Bayesian and gradient-descent trained neural networks converge to Gaussian Processes (GP) as the number of hidden units in intermediary layers approaches infinity (see §5). These results hold for many different architectures, including convolutional networks, which converge to a particular GP in the limit of infinite channels [Novak et al., 2019, Garriga-Alonso et al., 2019, Arora et al., 2019]. Bayesian networks are described by the Neural Network Gaussian Process (NNGP) kernel, while gradient descent networks are described by the Neural Tangent Kernel (NTK). Since we are interested in synthesizing datasets that can be used with both kernel methods and common gradient-descent trained neural networks, we focus on NTK in this work.

Infinitely wide networks have been shown to achieve SOTA (among non-parametric kernels) results on image classification tasks [Novak et al., 2019, Arora et al., 2019, Li et al., 2019, Shankar et al., 2020, Bietti, 2021] and even rival finite-width networks in certain settings [Arora et al., 2020, Lee et al., 2020]. This makes such kernels especially suitable for our task. As convolutional models, they encode useful inductive biases of locality and translation invariance [Novak et al., 2019], which enable good generalization. Moreover, flexible and efficient computation of these kernels are possible due to the Neural Tangent library [Novak et al., 2020].

**Specific models considered.** The central neural network (and corresponding infinite-width model) we consider is a simple 4-layer convolutional network with average pooling layers that we refer to as **ConvNet** throughout the text. This architecture is a slightly modified version of the default model used by Zhao and Bilen [2021], Zhao et al. [2021] and was chosen for ease of baselining (see §A for details). In several other settings we also consider convolutional networks without pooling

layers **ConvVec**,[2] and networks with no convolutions and only fully-connected layers **FC**. Depth of architecture (as measured by number of hidden layers) is indicated by an integer suffix.

**Background on algorithms.** We review the Kernel Inducing Points (KIP) and Label Solve (LS) algorithms introduced by Nguyen et al. [2021]. Given a kernel $K$, the kernel ridge-regression (KRR) loss function trained on a support dataset $(X_s, y_s)$ and evaluated on a target dataset $(X_t, y_t)$ is

$$L(X_s, y_s) = \frac{1}{2} \left\| y_t - K_{X_t X_s} (K_{X_s X_s} + \lambda I)^{-1} y_s \right\|_2^2, \tag{1}$$

where if $U$ and $V$ are sets, $K_{UV}$ is the matrix of kernel elements $(K(u, v))_{u \in U, v \in V}$. Here $\lambda > 0$ is a fixed regularization parameter. The KIP algorithm consists of minimizing (1) with respect to the support set (either just the $X_s$ or along with the labels $y_s$). Here, we sample $(X_t, y_t)$ from a target dataset $\mathcal{D}$ at every (meta)step, and update the support set using gradient-based methods. Additional variations include augmenting the $X_t$ or sampling a different kernel $K$ (from a fixed family of kernels) at each step.

The Label Solve algorithm consists of solving for the least-norm minimizer of (1) with respect to $y_s$. This yields the labels

$$y_s^* = \left( K_{X_t X_s} (K_{X_s X_s} + \lambda I)^{-1} \right)^+ y_t, \tag{2}$$

where $A^+$ denotes the pseudo-inverse of the matrix $A$. Note that here $(X_t, y_t) = \mathcal{D}$, i.e. the labels are solved using the whole target set.

In our applications of KIP and Label Solve, the target dataset $\mathcal{D}$ is always significantly larger than the support set $(X_s, y_s)$. Hence, the learned support set or solved labels can be regarded as distilled versions of their respective targets. We also initialize our support images to be a subset of natural images, though they could also be initialized randomly.

Based on the infinite-width correspondence outlined above and in §5, dataset distillation using KIP or LS that is optimized for KRR should extend to the corresponding finite-width neural network training. Our experimental results in §3 validate this expectation across many settings.

## 2.1 Client-Server Distributed Workflow

We invoke a client-server model of distributed computation[3], in which a server distributes independent workloads to a large pool of client workers that share a queue for receiving and sending work. Our distributed implementation of the KIP algorithm has two distinct stages:

**Forward pass:** In this step, we compute the support-support and target-support matrices $K(X_s, X_s)$ and $K(X_t, X_s)$. To do so, we partition $X_s \times X_s$ and $X_t \times X_s$ into pairs of images $(x, x')$, each with batch size $B$. We send such pairs to workers compute the respective matrix block $K(x, x')$. The server aggregates all these blocks to obtain the $K(X_s, X_s)$ and $K(X_t, X_s)$ matrices.

**Backward pass:** In this step, we need to compute the gradient of the loss $L$ (1) with respect to the support set $X_s$. We need only consider $\partial L / \partial X_s$ since $\partial L / \partial y_s$ is cheap to compute. By the chain rule, we can write

$$\frac{\partial L}{\partial X_s} = \frac{\partial L}{\partial (K(X_s, X_s))} \frac{\partial K(X_s, X_s)}{\partial X_s} + \frac{\partial L}{\partial (K(X_t, X_s))} \frac{\partial K(X_t, X_s)}{\partial X_s}.$$

The derivatives of $L$ with respect to the kernel matrices are inexpensive, since $L$ depends in a simple way on them (matrix multiplication and inversion). What is expensive to compute is the derivative of the kernel matrices with respect to the inputs. Each kernel element is an independent function of the inputs and a naive computation of the derivative of a block would require forward-mode differentiation, infeasible due to the size of the input images and the cost to compute the individual kernel elements. Thus our main novelty is to divide up the gradient computation into backward differentiation sub-computations, specifically by using the built-in function `jax.vjp` in JAX [Bradbury et al., 2018]. Denoting $K = K(X_s, X_s)$ or $K(X_t, X_s)$ for short-hand, we divide the

---

[2]The abbreviation "Vec" stands for vectorization, to indicate that activations of the network are vectorized (flattened) before the top fully-connected layer instead of being pooled.

[3]Implemented using Courier available at `https://github.com/deepmind/launchpad`.

Table 1: **Comparison with other methods.** The left group consists of neural network based methods. The right group consists of kernel ridge-regression. All settings for KIP involve the use of label-learning. Grayscale datasets use standard channel-wise preprocessing while RGB datasets use regularized ZCA preprocessing.

| | Imgs/ Class | DC[1] | DSA[1] | KIP FC[1] aug | LS ConvNet[2,3] | KIP ConvNet[2] no aug | aug |
|---|---|---|---|---|---|---|---|
| MNIST | 1 | 91.7±0.5 | 88.7±0.6 | 85.5±0.1 | 73.4 | **97.3±0.1** | **96.5±0.1** |
| | 10 | 97.4±0.2 | 97.8±0.1 | 97.2±0.2 | 96.4 | **99.1±0.1** | **99.1±0.1** |
| | 50 | 98.8±0.1 | 99.2±0.1 | 98.4±0.1 | 98.3 | **99.4±0.1** | **99.5±0.1** |
| Fashion- MNIST | 1 | 70.5±0.6 | 70.6±0.6 | - | 65.3 | **82.9±0.2** | 76.7±0.2 |
| | 10 | 82.3±0.4 | 84.6±0.3 | - | 80.8 | **91.0±0.1** | 88.8±0.1 |
| | 50 | 83.6±0.4 | 88.7±0.2 | - | 86.9 | **92.4±0.1** | 91.0±0.1 |
| SVHN | 1 | 31.2±1.4 | 27.5±1.4 | - | 23.9 | 62.4±0.2 | **64.3±0.4** |
| | 10 | 76.1±0.6 | 79.2±0.5 | - | 52.8 | 79.3±0.1 | **81.1±0.5** |
| | 50 | 82.3±0.3 | **84.4±0.4** | - | 76.8 | 82.0±0.1 | **84.3±0.1** |
| CIFAR-10 | 1 | 28.3±0.5 | 28.8±0.7 | 40.5±0.4 | 26.1 | **64.7±0.2** | 63.4±0.1 |
| | 10 | 44.9±0.5 | 52.1±0.5 | 53.1±0.5 | 53.6 | **75.6±0.2** | 75.5±0.1 |
| | 50 | 53.9±0.5 | 60.6±0.5 | 58.6±0.4 | 65.9 | 78.2±0.2 | **80.6±0.1** |
| CIFAR-100 | 1 | 12.8±0.3 | 13.9±0.3 | - | 23.8 | **34.9±0.1** | 33.3±0.3 |
| | 10 | 25.2±0.3 | 32.3±0.3 | - | 39.2 | 47.9±0.2 | **49.5±0.3** |

[1] DC [Zhao et al., 2021], DSA [Zhao and Bilen, 2021], KIP FC [Nguyen et al., 2021].

[2] Ours.

[3] LD [Bohdal et al., 2020] is another baseline which distills only labels using the AlexNet architecture. Our LS achieves higher test accuracy than theirs in every dataset category.

matrix $\partial L/\partial K$, computed on the server, into $B \times B$ blocks corresponding to $\partial L/\partial K(x, x')$, where $x$ and $x'$ each have batch size $B$. We send each such block, along with the corresponding block of image data $(x, x')$, to a worker. The worker then treats the $\partial L/\partial K(x, x')$ it receives as the cotangent vector argument of `jax.vjp` that, via contraction, converts the derivative of $K(x, x')$ with respect to $x$ into a scalar. The server aggregates all these partial gradient computations performed by the workers, over all possible $B \times B$ blocks, to compute the total gradient $\partial L/\partial X_s$ used to update $X_s$.

## 3 Experimental Results

### 3.1 Kernel Distillation Results

We apply the KIP and LS algorithms using the ConvNet architecture on the datasets MNIST [LeCun et al., 2010], Fashion MNIST [Xiao et al., 2017], SVHN [Netzer et al., 2011], CIFAR-10 [Krizhevsky, 2009], and CIFAR-100. Here, the goal is to condense the train dataset down to a learned dataset of size 1, 10, or 50 images per class. We consider a variety of hyperparameter settings (image preprocessing method, whether to augment target data, and whether to train the support labels for KIP), the full details of which are described in §A. For space reasons, we show a subset of our results in Table 1, with results corresponding to the remaining set of hyperparameters left to Tables A3-A10. We highlight here that a crucial ingredient for our strong results in the RGB dataset setting is the use of regularized ZCA preprocessing. Note the variable effect that our augmentations have on performance (see the last two columns of Table 1): they typically only provides a benefit for a sufficiently large support set. This result is consistent with Zhao et al. [2021], Zhao and Bilen [2021], in which gains from augmentations are also generally obtained from larger support sets. We tried varying the fraction (0.25 and 0.5) of each target batch that is augmented at each training step and found that while 10 images still did best without augmentations, 100 and 500 images typically did slightly better with some partial augmentations (versus none or full). For instance, for 500 images on CIFAR-10, we obtained 81.1% test accuracy using augmentation rate 0.5. Thus, our observation is that given that larger support sets can distill larger target datasets, as the former increases in size, the latter can be augmented more aggressively for obtaining optimal generalization performance.

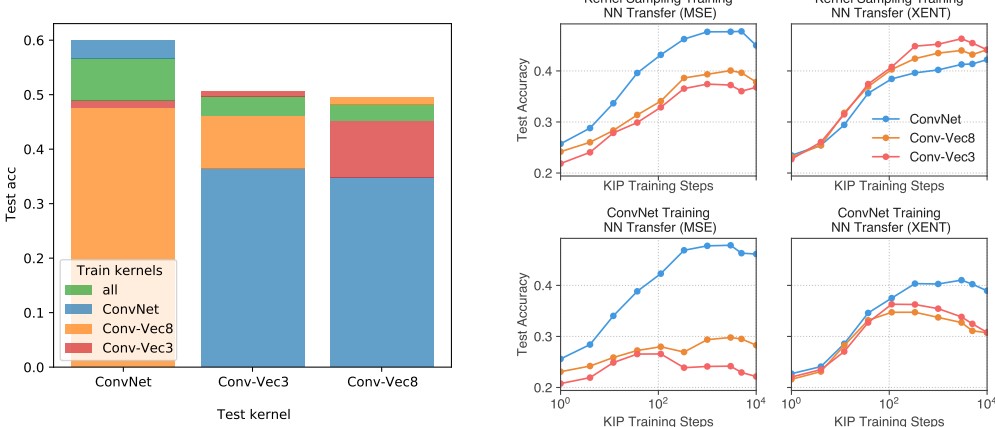

Figure 1: **KIP with kernel sampling vs individual kernels.** *Left:* Evaluation of three kernels, ConvNet, Conv-Vec3, Conv-Vec8 for KRR with respect to four train settings: sampling KIP ("all") which uses all the kernels or else KIP trained with the individual kernels. For all three kernels, "all" is a close second place, outperformed only if the kernel used for training is exactly the same as the one used for testing. *Right:* We take the learned images of the four train settings described above and transfer them to finite-width neural networks corresponding to ConvNet, Conv-Vec3, Conv-Vec8. Each point is a neural network trained on a specified KIP learned checkpoint. Top row is sampling KIP images and bottom row is the baseline using just ConvNet for KIP. These plots indicate that sampling KIP improves performance across the architectures that are sampled, for both MSE and cross entropy loss. Settings: CIFAR-10, 100 images, no augmentations, no ZCA, no label learning.

Remarkably, our results in Table 1 outperform all prior baselines across all dataset settings. Our results are especially strong in the small support size regime, with our 1 image per class results for KRR outperforming over 100 times as many natural images (see Table A1). We also obtain a significant margin over prior art across all datasets, with our largest margin being a 37% absolute gain in test accuracy for the SVHN dataset.

## 3.2  Kernel Transfer

In §3.1, we focused on obtaining state-of-the-art dataset distillation results for image classification using a specific kernel (ConvNet). Here, we consider the variation of KIP in which we sample from a family of kernels (which we call sampling KIP). We validate that sampling KIP adds robustness to the learned images in that they perform well for the family of kernels sampled during training.

In Figure 1, we plot test performance of sampling KIP when using the kernels ConvNet, Conv-Vec3, and Conv-Vec8 (denoted by "all") alongside KIP trained with just the individual kernels. Sampling KIP performs well at test time when using any of the three kernels, whereas datasets trained using a single kernel have a significant performance drop when using a different kernel.

## 3.3  Neural Network Transfer

In this section, we study how our distilled datasets optimized using KIP and LS transfer to the setting of finite-width neural networks. The main results are shown in Table 2. The third column shows the best neural network performance obtained by training on a KIP dataset of the corresponding size with respect to some choice of KIP and neural network training hyperparameters (see §A for details). Since the datasets are optimized for kernel ridge-regression and not for neural network training itself, we expect some performance loss when transferring to finite-width networks, which we record in the fourth column. Remarkably, the drop due to this transfer is quite moderate or small and sometimes the transfer can even lead to gain in performance (see LS for SVHN dataset with 10 images per class).

Overall, our transfer to finite-width networks outperforms prior art based on DC/DSA [Zhao et al., 2021, Zhao and Bilen, 2021] in the 1 image per class setting for all the RGB datasets (SVHN, CIFAR-10, CIFAR-100). Moreover, for CIFAR-10, we outperform DC/DSA in all settings.

Table 2: **Transfer of KIP and LS to neural network training.** Datasets obtained from KIP and LS using the ConvNet kernel are optimized for kernel ridge-regression and thus have reduced performance when used for training the corresponding finite-width ConvNet neural network. Remarkably, the loss in performance is mostly moderate and even small in many instances. Grayscale datasets use standard channel-wise preprocessing while RGB datasets use regularized ZCA preprocessing. The KIP datasets used here can have augmentations or no augmentations and, unlike those in Table 1, can have either fixed or learned labels. $*$ denotes best chosen transfer is obtained with learned labels.

|  | Imgs/Class | DC/DSA | KIP to NN | Perf. change | LS to NN | Perf. change |
|---|---|---|---|---|---|---|
| MNIST | 1 | **91.7±0.5** | 90.1±0.1 | -5.5 | 71.0±0.2 | -2.4 |
|  | 10 | **97.8±0.1** | 97.5±0.0 | -1.1 | 95.2±0.1 | -1.2 |
|  | 50 | **99.2±0.1** | 98.3±0.1 | -0.8 | 97.9±0.0 | -0.4 |
| Fashion-MNIST | 1 | 70.6±0.6 | **73.5±0.5**$^*$ | -9.8 | 61.2±0.1 | -4.1 |
|  | 10 | 84.6±0.3 | **86.8±0.1** | -1.3 | 79.7±0.1 | -1.2 |
|  | 50 | **88.7±0.2** | 88.0±0.1$^*$ | -4.5 | 85.0±0.1 | -1.8 |
| SVHN | 1 | 31.2±1.4 | **57.3±0.1**$^*$ | -8.3 | 23.8±0.2 | -0.2 |
|  | 10 | **79.2±0.5** | 75.0±0.1 | -1.6 | 53.2±0.3 | 0.4 |
|  | 50 | **84.4±0.4** | 80.5±0.1 | -1.0 | 76.5±0.3 | -0.4 |
| CIFAR-10 | 1 | 28.8±0.7 | **49.9±0.2** | -9.2 | 24.7±0.1 | -1.4 |
|  | 10 | 52.1±0.5 | **62.7±0.3** | -4.6 | 49.3±0.1 | -4.3 |
|  | 50 | 60.6±0.5 | **68.6±0.2** | -4.5 | 62.0±0.2 | -3.9 |
| CIFAR-100 | 1 | 13.9±0.3 | **15.7±0.2**$^*$ | -18.1 | 11.8±0.2 | -12.0 |
|  | 10 | **32.3±0.3** | 28.3±0.1 | -17.4 | 25.0±0.1 | -14.2 |

Figures 2 and 3 provide a closer look at KIP transfer changes under various settings. The first of these tracks how transfer performance changes when adding additional layers as function of the number of KIP training steps used. The normalization layers appear to harm performance for MSE loss, which can be anticipated from their absence in the KIP and LS optimization procedures. However they appear to provide some benefit for cross entropy loss. For Figure 3, we observe that as KIP training progresses, the downstream finite-width network's performance also improves in general. A notable exception is observed when learning the labels in KIP, where longer training steps lead to deterioration of information useful to training finite-width neural networks. We also observe that as predicted by infinite-width theory [Jacot et al., 2018, Lee et al., 2019], the overall gap between KIP or LS performance and finite-width neural network decreases as the width increases. While our best performing transfer is obtained with width 1024, Figure 3 (middle) suggest that even with modest width of 64, our transfer can outperform prior art of 60.6% by Zhao and Bilen [2021].

Finally, in Figure 4, we investigate the performance of KIP images over the course of training of a single run, as compared to natural images, over a range of hyperparameters. We find the outperformance of KIP images above natural images consistent across hyperparameters and checkpoints. This suggests that our KIP images may also be effective for accelerated hyperparameter search, an application of dataset distillation explored in Zhao et al. [2021], Zhao and Bilen [2021].

Altogether, we find our neural network training results encouraging. First, it validates the applicability of infinite-width methods to the setting of finite width [Huang and Yau, 2020, Dyer and Gur-Ari, 2020, Andreassen and Dyer, 2020, Yaida, 2020, Lee et al., 2020]. Second, we find some of the transfer results quite surprising, including efficacy of label solve and the use of cross-entropy for certain settings (see §A for the full details).

## 4 Understanding KIP Images and Labels

A natural question to consider is what causes KIP to improve generalization performance. Does it simplify support images, removing noise and minor sources of variation while keeping only the core features shared between many target images of a given class? Or does it make them more complex by producing outputs that combine characteristics of many samples in a single resulting collage? While

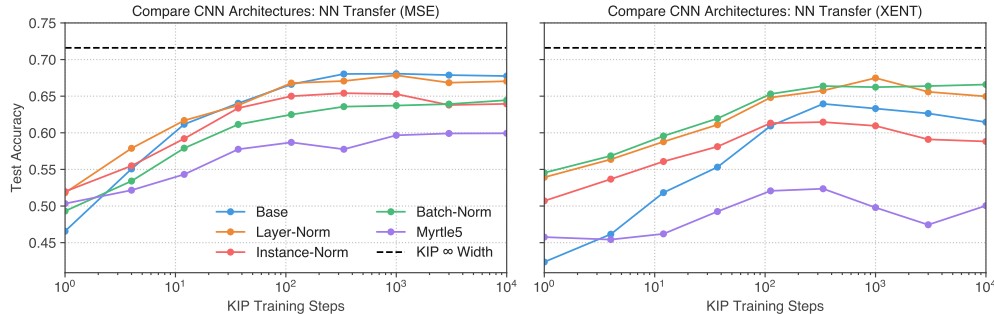

Figure 2: **Robustness to neural network variations.** KIP ConvNet images (trained with fixed labels) are tested on variations of the ConvNet neural network, including those which have various normalization layers (layer, instance, batch). A similar architecture to ConvNet, the Myrtle5 architecture (without normalization layers) [Shankar et al., 2020], which differs from the ConvNet architecture by having an additional convolutional layer at the bottom and a global average pooling that replaces the final local average pooling at the top, is also tested. Finally, mean-square error is compared with cross-entropy loss (left versus right). Settings: CIFAR-10, 500 images, ZCA, no label learning.

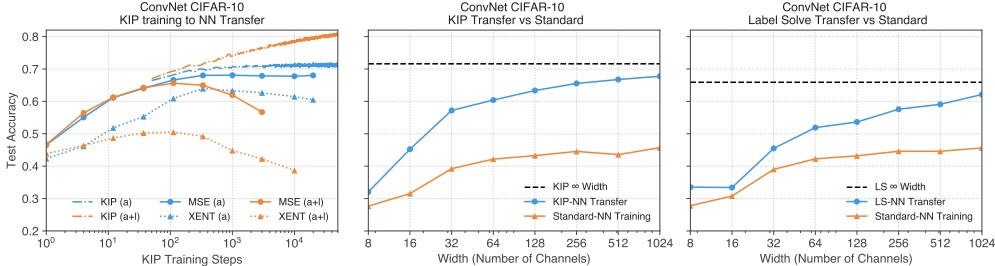

Figure 3: **Variations for neural network transfer.** *Left:* Plot of transfer performance as a function of KIP training steps across various train settings. Here (a) denotes augmentations used during KIP training and (a+l) denotes that additionally the labels were learned. MSE and XENT denote mean-square-error and cross entropy loss for the neural network, where for the case of XENT and (a+l), the labels for the neural network are the argmax of the learned labels. *Middle:* Exploring the effect of width on transferability of vanilla KIP data. *Right:* The effect of width on the transferability of label solved data. Settings: CIFAR-10, 500 images, ZCA.

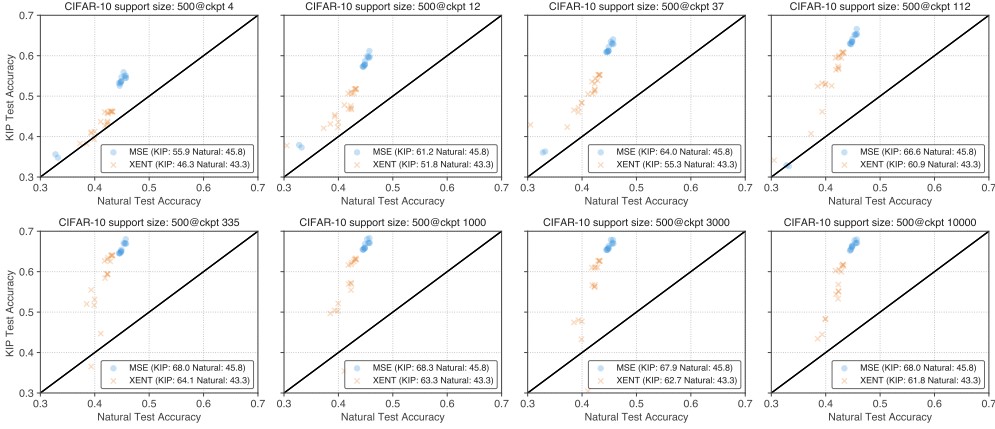

Figure 4: **Hyperparameter robustness.** In the above, KIP images across eight different checkpoints are used to train the ConvNet neural network. Each point in each plot is a neural network training with a different hyperparameter, and its location records the final test accuracy when training on natural images versus the KIP images obtained from initializing from such images. For both MSE and cross entropy loss, KIP images consistently exceed natural images across many hyperparameters. Settings: CIFAR-10, 500 images, ZCA, no augmentations, no label learning.

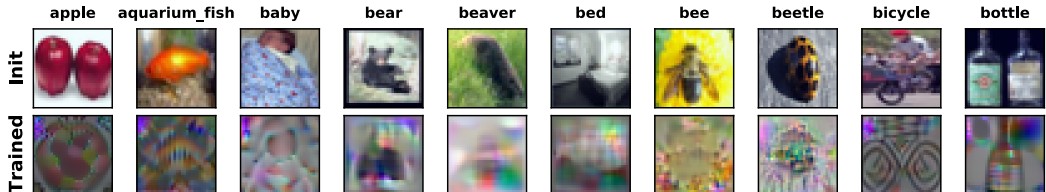

Figure 5: **Examples of learned images.** Images are initialized from natural images in the top row (`Init`) and converge to images in the bottom row (`Trained`). Settings: 100 images distilled, no ZCA, no label training, augmentations.

properly answering this question is subject to precise definitions of simplicity and complexity, we find that KIP tends to increase the complexity of pictures based on the following experiments:

- **Visual analysis**: qualitatively, KIP images tend to be richer in textures and contours.
- **Dimensional analysis**: KIP produces images of higher intrinsic dimensionality (i.e. an estimate of the dimensionality of the manifold on which the images lie) than natural images.
- **Spectral analysis**: unlike natural images, for which the bulk of generalization performance is explained by a small number of top few eigendirections, KIP images leverage the whole spectrum much more evenly.

Combined, these results let us conclude that KIP usually increases complexity, integrating features from many target images into much fewer support images.

**Visual Analysis.** A visual inspection of our learned data leads to intriguing observations in terms of interpretability. Figure 5 shows examples of KIP learned images from CIFAR-100. The resulting images are heterogeneous in terms of how they can be interpreted as distilling the data. For instance, the distilled apple image seems to consist of many apples nested within a possibly larger apple, whereas the distilled bottle image starts off as two bottles and before transforming into one, while other classes (like the beaver) are altogether visually indistinct. Investigating and quantifying aspects that make these images generalize so well is a promising avenue for future work. We show examples from other datasets in Figure A2.

In Figure 6, we compare MNIST KIP data learned with and without label learning. For the latter case with images and labels optimized jointly, while labels become more informative, encoding richer inter-class information, the images become less interpretable. This behavior consistently leads to superior KRR results, but appears to not be leveraged as efficiently in the neural network transfer setting (Table 2). Experimental details can be found in §A.

**Dimensional Analysis.** We study the intrinsic dimension of KIP images and find that they tend to grow. Intrinsic dimension (ID) was first defined by Bennett [1969] as "the number of free parameters required in a hypothetical signal generator capable of producing a close approximation to each signal in the collection". In our context, it is the dimensionality of the manifold embedded into the image space which contains all the support images. Intuitively, simple datasets have a low ID as they can be described by a small number of coordinates on the low-dimensional data manifold.

ID can be defined and estimated differently based on assumptions on the manifold structure and the probability density function of the images on this manifold (see [Camastra and Staiano, 2016] for review). We use the "Two-NN" method developed by [Facco et al., 2017], which makes relatively few assumptions and allows to estimate ID only from two nearest-neighbor distances for each datapoint.

Figure 7 shows that the ID is increasing for the learned KIP images as a function of the training step across a variety of configurations (training with or without augmentations/label learning) and datasets. One might expect that a distillation procedure should decrease dimensionality. On the other hand, Ansuini et al. [2019] showed that ID increases in the earlier layers of a trained neural network. It remains to be understood if this latter observation has any relationship with our increased ID. Note that ZCA preprocessing, which played an important role for getting the best performance for our RGB datasets, increases dimensionality of the underlying data (see Figure A4).

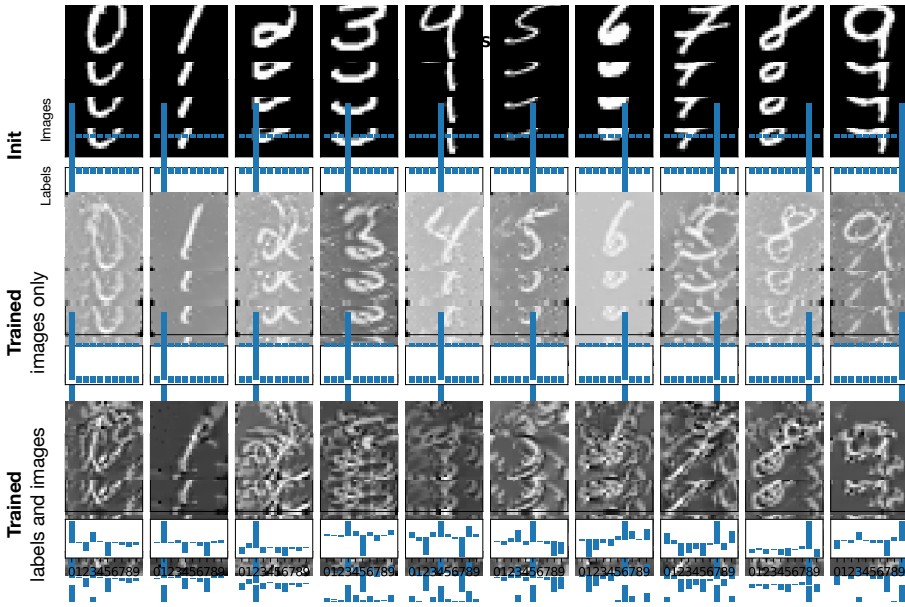

Figure 6: **Dataset distillation with trainable and non-trainable labels.** *Top row:* initialization of support images and labels. *Middle row:* trained images if labels remain fixed. *Bottom row:* trained images and labels, jointly optimized. Settings: 100 images distilled, no augmentations.

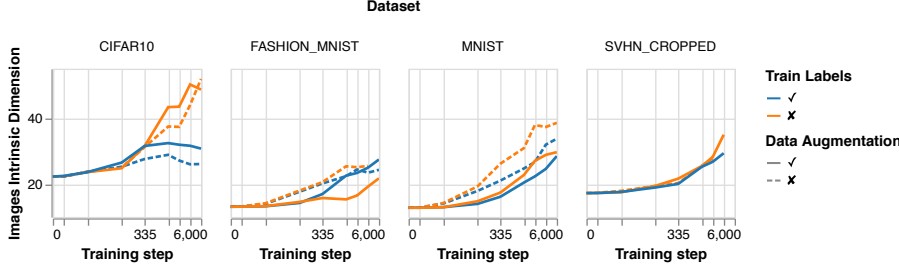

Figure 7: **Intrinsic dimension of a learned datasets grows during training.** As training progresses, intrinsic dimension of the learned dataset grows, indicating that training non-trivially transforms the data manifold. See Figures A2 and 6 for visual examples of learned images, and Figures A3 and A4 for similar observations using other metrics and settings. Settings: 500 images distilled, no ZCA.

**Spectral Analysis.** Another distinguishing property of KIP images is how their spectral components contribute to performance. In Figure 8, we spot how different spectral bands of KIP learned images affect test performance as compared to their initial natural images. Here, we use the FC2, Conv-Vec8, and ConvNet architectures. We note that for natural images (light bars), most of their performance is captured by the top 20% of eigenvalues. For KIP images, the performance is either more evenly distributed across the bands (FC and Conv-Vec8) or else is skewed towards the tail (ConvNet).

## 5 Related Work

Dataset distillation was first studied in Wang et al. [2018]. The work of Sucholutsky and Schonlau [2019], Bohdal et al. [2020] build upon it by distilling labels. Zhao et al. [2021] proposes condensing a training set by harnessing a gradient matching condition. Zhao and Bilen [2021] takes this idea further by applying a suitable augmentation strategy. Note that while Zhao and Bilen [2021] is limited in augmentation expressiveness (they have to apply a single augmentation per training iteration), we can sample augmentations independently per image in our target set per train step. Our work together with Nguyen et al. [2021] are, to the best of our knowledge, the only works using kernel-based methods for dataset distillation on image classification datasets.

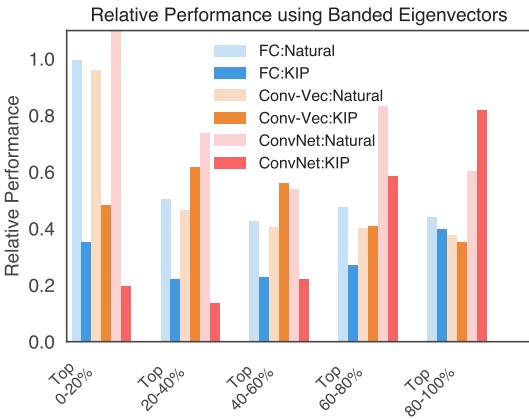

Figure 8: **Spectral contribution to test accuracy shifts to the tail.** By setting the ridge-parameter to zero in kernel-ridge regression and composing $K_{X_s X_s}^{-1}$ with the spectral projection onto various eigenspaces, we can explore how different spectral bands affect test accuracy of kernel ridge-regression. We plot the relative change in test accuracy using contiguous bands of 20% of the eigenvalues. Settings: CIFAR-10, 500 images. Further details in §A.

Our use of kernels stems from the correspondence between infinitely-wide neural networks and kernel methods [Neal, 1994, Lee et al., 2018, Matthews et al., 2018, Jacot et al., 2018, Novak et al., 2019, Garriga-Alonso et al., 2019, Arora et al., 2019, Yang, 2019a,b, Hron et al., 2020] and the extended correspondence with the finite width corrections [Dyer and Gur-Ari, 2020, Huang and Yau, 2020, Yaida, 2020]. These correspondences underlie the transferability of our KRR results to neural networks , and have been utilized in understanding trainability [Xiao et al., 2020], generalizations [Adlam and Pennington, 2020], training dynamics [Lewkowycz et al., 2020, Lewkowycz and Gur-Ari, 2020], uncertainty [Adlam et al., 2021], and demonstrated their effectiveness for smaller datasets [Arora et al., 2020] and neural architecture search [Park et al., 2020, Chen et al., 2021].

## 6 Conclusion

We performed an extensive study of dataset distillation using the KIP and LS algorithms applied to convolutional architectures, obtaining SOTA results on a variety of image classification datasets. In some cases, our learned datasets were more effective than a natural dataset two orders of magnitude larger in size. There are many interesting followup directions and questions from our work:

First, integrating efficient kernel-approximation methods into our algorithms, such as those of Zandieh et al. [2021], will reduce computational burden and enable scaling up to larger datasets. In this direction, the understanding of how various resources (e.g. data, parameter count, compute) scale when optimizing for neural network performance has received significant attention as machine learning models continue to stretch computational limits [Hestness et al., 2017, Rosenfeld et al., 2020, Kaplan et al., 2020, Bahri et al., 2021]. Developing our understanding of how to harness smaller, yet more useful representations data would aid in such endeavors. In particular, it would be especially interesting to explore how well datasets can be compressed as they scale up in size.

Second, LS and KIP with label learning shows that optimizing labels is a very powerful tool for dataset distillation. The labels we obtain are quite far away from standard, interpretable labels and we feel their effectiveness suggests that understanding of how to optimally label data warrants further study.

Finally, the novel features obtained by our learned datasets, and those of dataset distillation methods in general, may reveal insights into interpretability and the nature of sample-efficient representations. For instance, observe the `bee` and `bicycle` images in Figure 5: the `bee` class distills into what appears to be spurious visual features (e.g. pollen), while the `bicycle` class distills to the essential contours of a typical bicycle. Additional analyses and explorations of this type could offer insights into the perennial question of how neural networks learn and generalize.

**Acknowledgments** We would like to acknowledge special thanks to Samuel S. Schoenholz, who proposed and helped develop the overall strategy for our distributed KIP learning methodology. We are also grateful to Ekin Dogus Cubuk and Manuel Kroiss for helpful discussions.

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
