# A Experimental Details

We provide the details of the various configurations in our experiments:

- **Augmentations**: For RGB datasets, we apply learned policy found in the AutoAugment [Cubuk et al., 2019] scheme, followed by horizontal flips. We found crops and cutout to harm performance in our experiments. For grayscale datasets, we implemented augmentations in Keras's `tf.keras.preprocessing.image.ImageDataGenerator` class, using rotations up to 10 degrees and height and width shift range of 0.1.

- **Datasets**: We use the MNIST, Fashion-MNIST, CIFAR-10, CIFAR-100, and SVHN (cropped) datasets as provided by the `tensorflow_datasets` library. When using mean-square error loss, labels are (mean-centered) one-hot labels.

- **Initialization**: We initialize KIP and LS with class-balanced subsets of the train data (with the corresponding preprocessing). One could also initialize with uniform random noise. We found the final performance and visual quality of the learned images to be similar in either case.

- **Preprocessing**: We consider two preprocessing schemes. By default, there is *standard preprocessing*, in which channel-wise mean and standard deviation are computed across the train dataset and then used to normalize the train and test data. We also have *(regularized) ZCA*[4], which depends on a regularization parameter $\lambda \geq 0$ as follows. First, we flatten the features for each train image and then standardize each feature across the train dataset. The feature-feature covariance matix $C$ has a singular value decomposition $C = U\Sigma U^T$, from which we get a regularized whitening matrix given by $W_\lambda = U\phi_\lambda(\Sigma)U^T$, where $\phi_\lambda$ maps each singular-value $\mu$ to $(\mu + \lambda\overline{\mathrm{tr}}C)^{-1/2}$ and where

$$\overline{\mathrm{tr}}(C) = \mathrm{tr}(C)/\mathrm{len}(C). \qquad (3)$$

  Then regularized ZCA is the transformation which applies layer normalization to each flattened feature vector (i.e. standardize each feature vector along the feature axis) and then multiplies the resulting feature by $W_\lambda$. (The case $\lambda = 0$ is ordinary ZCA preprocessing). Our implementation of regularized ZCA preprocessing follows that of Shankar et al. [2020]. We apply regularized ZCA preprocessing for RGB datasets, with $\lambda = 100$ for SVHN and 0.1 for CIFAR-10 and CIFAR-100 unless stated otherwise. We obtained these values by tuning the regularization strength on a 5K subset of training and validation data. If a dataset has no ZCA preprocessing, then it has standard preprocessing.

- **Label training**: This is a boolean hyperparameter in all our KIP training experiments. When true, we optimize the support labels $y_s$ in the KIP algorithm. When false, they remain fixed at their initial (centered one-hot) values.

Other experimental details include the following. We use NTK parameterization[5] for exact computation of infinite-width neural tangent kernels. For KIP training, we use target batch size of 5K (the number of elements of $(X_t, y_t)$ in (1)) and train for up to 50K iterations[6] (some runs we end early due to training budget or because test performance saturated). We used the Adam optimizer with learning rate 0.04 for CIFAR-10 and CIFAR-100, and 0.01 for the remaining datasets unless otherwise stated. Our kernel regularizer $\lambda$ in (1), instead of being a fixed constant, is adapted to the scale of $K_{X_s X_s}$, namely

$$\lambda = \lambda_0 \overline{\mathrm{tr}}(K_{X_s X_s})$$

where $\lambda_0 = 10^{-6}$.

**ConvNet Architecture.** In Zhao et al. [2021], Zhao and Bilen [2021], the ConvNet architecture used consists of three blocks of convolution, instance normalization, ReLu, and 2x2 average pooling, followed by a linear readout layer. In our work, we remove the instance normalization (in order to invoke the infinite-width limit) and we additionally prepend an additional convolution and ReLu layer at the beginning of the network, making our network consist of 4 convolutional layers instead of 3.[7] The two architectures are comparable in terms of performance, see §C.4 and Tables A12 and A12.

**Table 1**: All KIP runs use label training and for both KIP and LS, regularized ZCA preprocessing is used for RGB datasets. All KIP test accuracies have mean and standard deviation computed with respect to 5 checkpoints based on 5 lowest train loss checkpoints (we checkpoint every 50 train steps, for which a target batch size of

---

[4]Our approach is consistent with that of Shankar et al. [2020].

[5]Improved standard parameterization of NTK is described in Sohl-Dickstein et al. [2020].

[6]In practice, most of the convergence is achieved after a thousand steps, with a slow, logarithmic increase in test performance with more iterations in many instances, usually when there is label training.

[7]The latter modification was unintentional, as this additional block also occurs in the Myrtle architecture used in other kernel based works e.g. Shankar et al. [2020], Lee et al. [2020], Nguyen et al. [2021].

5K means roughly every 5 epochs). Given the expensiveness of our runs (each of which requires hundreds of GPUs), it is prohibitively expensive to average runs over random seeds. In practice, we find very small variance between different runs with the same hyperparameters. The LS numbers, likewise, were computed with respect to a random support set from a single random seed. We use hyperparameters based on the previous discussion, except for the case of SVHN without augmentations. We found training to be unstable early on in this case, possibly due to the ZCA regularization of 100, while being optimal for kernel ridge-regression, leads to poor conditioning for the gradient updates in KIP. So for the case of SVHN without augmentations, we fall back to the CIFAR10 hyperparameters of ZCA regularization of 0.1 and learning rate of 0.04.

**Details for neural network transfer:** For neural network transfer experiments in Tables 2, A2, A5, A7, A9, Figures 1 (right), 2, 3, and 4, we follow training and optimization hyperparameter tuning scheme of Lee et al. [2020], Nguyen et al. [2021].

We trained the networks with cosine learning rate decay and momentum optimizer with momentum 0.9. Networks were trained using mean square (MSE) or softmax-cross-entropy (XENT) loss. We used full-batch gradient for support size 10 and 100 whereas for larger support sizes (500, 1000) batch-size of 100 was used.

KIP training checkpoint at steps $\{0, 1, 4, 12, 37, 112, 335, 1000, 3000, 10000, 20000, 50000\}$ were used for evaluating neural network transfer, with the best results being selected from among them. For each checkpoint transfer, initial learning rate and L2 regularization strength were tuned over a small grid search space. Following the procedure used in Lee et al. [2020], learning rate is parameterized with learning rate factor $c$ with respect to the critical learning rate $\eta = c\, \eta_{\text{critical}}$ where $\eta_{\text{critical}} \equiv 2/(\lambda_{\min}(\Theta) + \lambda_{\max}(\Theta))$ and $\Theta$ is the NTK [Lee et al., 2019]. In practice, we compute the empirical NTK $\hat{\Theta}(x, x') = \sum_j \partial_j f(x) \partial_j f(x')$ on `min(64, support_size)` random points in the training set to estimate $\eta_{\text{critical}}$ [Lee et al., 2019] by maximum eigenvalue of $\hat{\Theta}(x, x)$. Grid search was done over the range $c \in \{0.5, 1, 2, 4\}$ and similarly for the L2-regularization strength in the range $\{0, 10^{-7}, 10^{-5}, 10^{-3}, 10^{-1}\}$. The best hyperparameters and early stopping are based on performance with respect to a separate 5K validation set. From among all these options, the best result is what appears as an entry in the KIP to NN columns of our tables. The Perf. change column of Table 2 measures the difference in the neural network performance with the kernel ridge-regression performance of the chosen KIP data. The error bars when listed is standard deviation on top-20 measurements based on validation set.

The number of training steps are chosen to be sufficiently large. For most runs 5,000 steps were enough, however for CIFAR-100 with MSE loss we increased the number of steps to 25,000 and for SVHN we used 10,000 steps to ensure the networks can sufficiently fit the training data. In training full (45K/5K/10K split) CIFAR-10 training dataset in Table A2, we trained for 30,000 steps for XENT loss and 300,000 steps for MSE loss. When standard data augmentation (flip and crop) is used we increase the number of training steps for KIP images and full set of natural images to 50,000 and 500,000 respectively.

Except for the width/channel variation studies, 1,024 channel convolutional layers were used. The network was initialized and parameterized by standard parameterization instead of NTK parameterization. Data preprocessing matched that of corresponding KIP training procedure.

When transferring learned labels and using softmax-cross-entropy loss, we used the one-hot labels obtained by taking the maximum argument index of learned labels as true target class.

**Figures 7, A3, A4**: Intrinsic dimension is measured using the code provided by Ansuini et al. [2019][8] with default settings. Linear dimension is measured by the smallest number of PCA components needed to explain 90% of variance in the data. Gradient [linear] dimension is measured by considering the analytic, infinite-width NTK in place of the data covariance matrix, either for the purpose of computing PCA to establish the linear dimension, or for the purpose of producing the pairwise-distance between gradients. Precisely, for a finite-width neural network $f_n$ of width $n$ with trainable parameters $\theta$, we can use the parallelogram law to obtain

$$\left\| \frac{\partial f_n(x_1)}{\partial \theta} - \frac{\partial f_n(x_2)}{\partial \theta} \right\|_2^2 = \left\| \frac{\partial f_n(x_1)}{\partial \theta} \right\|_2^2 + \left\| \frac{\partial f_n(x_2)}{\partial \theta} \right\|_2^2 - 2 \left( \frac{\partial f_n(x_1)}{\partial \theta}^T \frac{\partial f_n(x_2)}{\partial \theta} \right) \tag{4}$$

$$= \hat{\Theta}_n(x_1, x_1) + \hat{\Theta}_n(x_2, x_2) - 2\hat{\Theta}_n(x_1, x_2) \xrightarrow[n \to \infty]{} \tag{5}$$

$$\Theta(x_1, x_1) + \Theta(x_2, x_2) - 2\Theta(x_1, x_2). \tag{6}$$

Therefore, NTK can be used to compute the limiting pairwise distance between gradients, and consequently the intrinsic dimension of the manifold of these gradients.

**Figure 8**: We mix different KIP hyperparameters for variety. ConvNet: ZCA, no label learning; Conv-Vec: no ZCA, no label learning, 8 hidden layers; FC: no ZCA, label learning, and 2 hidden layers. All three settings used augmentations and and target batch size of 50K (full gradient descent) instead of 5K.

---

[8] https://github.com/ansuini/IntrinsicDimDeep

**Figure A1**: The eight hyperparameters are given by dataset size (100 or 500), zca (+) or no zca (-), and train labels (+) or no train labels (-). All runs use no augmentations. The datasets were obtained after 1000 steps of training. Their ordering in terms of test accuracy (in ascending order) are given by (100, -, -), (100, -, +), (500, -, -), (100, +, -), (500, -, +), (100, +, +), (500, +, -), (500, +, +).

# B  Computational Costs

Classical kernels (RBF and Laplace) as well as FC kernels are cheap to compute in the sense that they are dot product kernels: a kernel matrix element $k(x, x')$ between two inputs only depends on $x \cdot x$, $x \cdot x'$, and $x' \cdot x'$. However, kernels formed out of convolutional and pooling layers introduce computational complexity by having to keep track of pixel-pixel correlations. Consequently, the addition of a convolutional layer introduces an $O(d)$ complexity while an additional (global) pooling layer introduces $O(d^2)$ complexity, where $d$ is the spatial dimension of the image (i.e. height times width).

As an illustration, when using the `neural_tangents` [Novak et al., 2020] library, one is able to use roughly the following kernel batch sizes[9] on a Tesla V100 GPU (16GB RAM):

$$
\begin{aligned}
\text{FC1 kernel batch size} \quad &\sim \quad 10^4 \\
\text{Conv-Vec1 kernel batch size} \quad &\sim \quad 10^3 \\
\text{ConvNet kernel batch size} \quad &\sim \quad 10^1.
\end{aligned}
$$

Thus, for the convolutional kernels considered in this paper, our KIP algorithm requires many parallel resources in order for training to happen at a reasonable speed. For instance, for a support set of size 100 and a target batch size of 5000, the $K(X_s, X_s)$ and $K(X_t, X_s)$ matrix that needs to be computed during each train step has roughly $5 \times 10^5$ many elements. A kernel batch size of $B$ means that we can only compute a $B \times B$ sub-block of $K(X_t, X_s)$ at a time. For $B = 18$ (the maximum batch size for ConvNet on a V100 GPU), this leads to $1.5 \times 10^3$ computations. With a cost of about 80 milliseconds to compute each block, this amounts to 120 seconds of work to compute the required kernel matrices per train step. Computing the gradients takes about three times longer, and so this amounts to 460 seconds of time per train step.

Our distributed framework as described in §2.1, using hundreds of GPUs working in parallel, is what enables our training to become computationally feasible.

# C  Additional Tables and Figures

## C.1  KIP Image Analysis

**Subsampling.** The train and test images from our benchmark datasets are sampled from the distribution of natural images. KIP images however are jointly learned and therefore correlated. In Figure A1 this is validated by showing that natural images degrade in performance (test accuracy using kernel ridge-regression) more gracefully than KIP images when subsampling.

**Visual analysis.** We provide analogous images to Figure 5 for the remaining datasets in Figure A2.

**Dimensional Analysis.** We provide some additional plots of dimension-based measures of our learned images in Figures A3, 7. In addition to the intrinsic dimension, we measure the linear dimension and the gradient based versions of the intrinsic and linear dimension (see Section A for details).

## C.2  Natural Image Baselines

We subsample natural image subsets of various sizes and evaluate test accuracy using kernel-ridge regression with respect to the ConvNet kernel. This gives a sense of how well KIP images perform relative to natural images. In particular, from Table 1, KIP images using only 1 image per class outperform over 100 images per class.

We also consider how ZCA preprocessing, the presence of instance normalization layers (which recovers the ConvNet used in Zhao et al. [2021], Zhao and Bilen [2021]), augmentations, and choice of loss function affect neural network training of ConvNet. We record the results with respect to CIFAR-10 in Table A2.

---

[9] A kernel matrix element $K(x, x')$ is a function of a pair of images $x, x' \in \mathbb{R}^{H \times W \times C}$. By vectorizing the computation, we can apply $K$ to batches of images $x, x' \in \mathbb{R}^{B \times H \times W \times C}$ to obtain a $B \times B$ matrix. We call $B$ the (kernel) batch size.

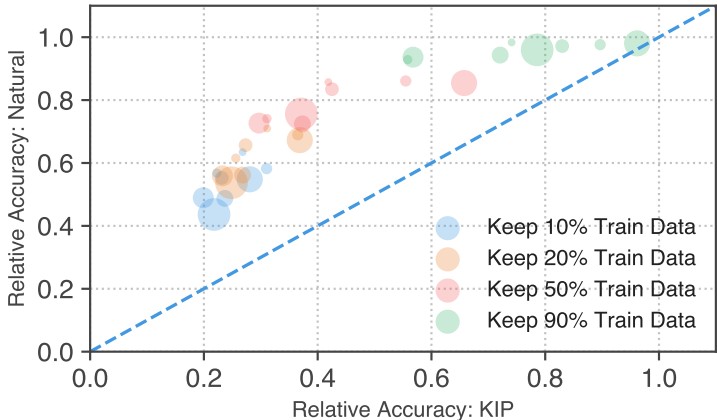

Figure A1: **KIP images are highly correlated.** Subsamples of KIP images are compared to the corresponding subsample of natural images at initialization. The relative drop in test accuracy among the two cases are plotted along the axes. KIP images show more severe degradation under subsampling. A selection of eight hyperparameters (see §A) were chosen among KIP trained images. Each hyperparameter is a differently sized circle, with larger circles indicating higher test accuracy when using full support set (i.e. keep 100% train data). Different colors denote different subset sizes, and test accuracies are computed using the average of 100 randomly sampled subsets. Settings: CIFAR-10.

Table A1: ConvNet kernel performance on random subsets. Mean and standard deviation computed over twenty random subsets. "All" row denotes entire dataset.

| Imgs/Class | MNIST | Fashion MNIST | SVHN | SVHN (ZCA) |
|---|---|---|---|---|
| 1 | 52.0±5.1 | 47.0±5.7 | 10.8±1.1 | 13.1±1.7 |
| 2 | 65.2±3.2 | 56.8±3.2 | 12.6±1.5 | 14.8±1.6 |
| 4 | 79.9±1.9 | 65.0±3.5 | 13.8±1.3 | 17.2±1.5 |
| 8 | 88.2±1.1 | 70.2±1.8 | 16.8±1.4 | 23.0±1.6 |
| 16 | 93.2±0.7 | 74.9±1.3 | 23.4±1.3 | 32.0±2.1 |
| 32 | 95.5±0.3 | 78.4±0.9 | 32.6±1.4 | 45.2±1.2 |
| 64 | 96.9±0.2 | 82.1±0.5 | 44.3±1.0 | 58.1±1.2 |
| 128 | 97.8±0.1 | 84.8±0.3 | 55.1±0.8 | 67.6±0.9 |
| All | 99.4 | 93.0 | 85.1 | 88.8 |

| Imgs/Class | CIFAR-10 | CIFAR-10 (ZCA) | CIFAR-100 | CIFAR-100 (ZCA) |
|---|---|---|---|---|
| 1 | 16.3±2.1 | 16.1±2.1 | 4.5±0.5 | 5.7±0.5 |
| 2 | 18.2±2.0 | 19.9±2.3 | 6.3±0.4 | 7.9±0.6 |
| 4 | 20.8±1.4 | 23.7±1.7 | 8.8±0.4 | 11.3±0.6 |
| 8 | 24.5±1.8 | 29.2±1.7 | 12.5±0.4 | 16.3±0.5 |
| 16 | 29.5±1.3 | 36.2±1.1 | 17.1±0.4 | 22.5±0.4 |
| 32 | 35.7±1.0 | 44.1±1.1 | 22.5±0.4 | 29.2±0.4 |
| 64 | 41.7±1.1 | 51.5±0.9 | 28.6±0.3 | 36.5±0.3 |
| 128 | 47.9±0.3 | 58.1±0.6 | 35.1±0.4 | 43.6±0.3 |
| All | 76.1 | 83.2 | 48.1 | 56.3 |

## C.3 Ablation Studies

We present a complete hyperparameter sweep for KIP and LS in Table A3-A10 by including results for standard preprocessing and no label training, which was not on display in Table 1. We also provide corresponding neural network transfer results where available. All hyperparameters are as described in §A. Overall, we observe a clear and consistent benefit of using ZCA regularization and label training for KIP.

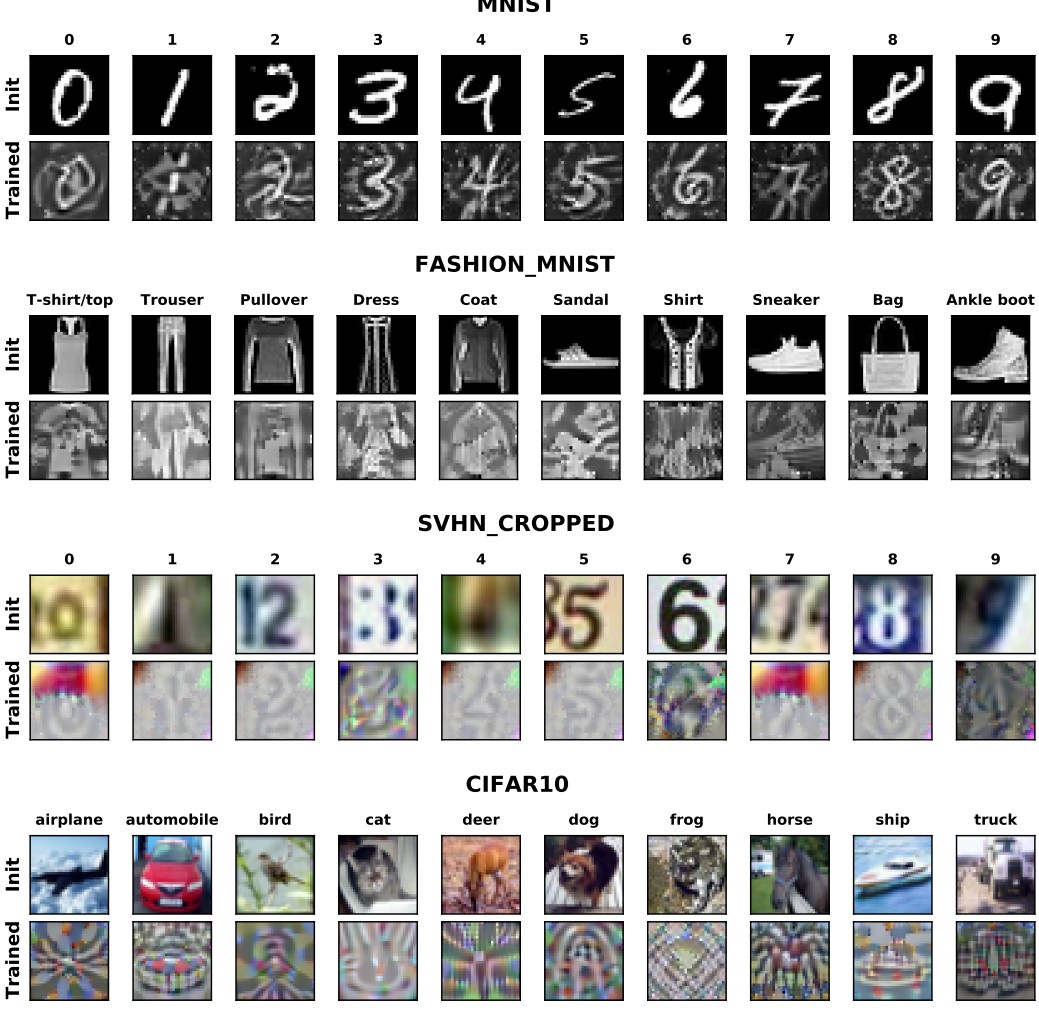

Figure A2: **Examples of learned images on more datasets.** See Figure 5 for CIFAR-100. Settings: 10 images distilled, no ZCA, no label training, no augmentations.

## C.4 DC/DSA ConvNet

As explained in Section §A, our ConvNet is a slightly modified version of the ConvNet appearing in the baselines Zhao et al. [2021], Zhao and Bilen [2021]. We compute kernel-ridge regression results for CIFAR-10 with the initial convolution and ReLu layer of our ConvNet removed in Tables A11 and A12. Since the architecture change is relatively minor, the numbers do not differ as much. Indeed, the differences are typically less than 1% (the largest difference was 1.9%), with the shallower ConvNet sometimes outperforming the deeper ConvNet. This suggests that our overall results are robust with respect to this change and hence are a fair comparison with the shallower 3-layer ConvNet of Zhao et al. [2021], Zhao and Bilen [2021].

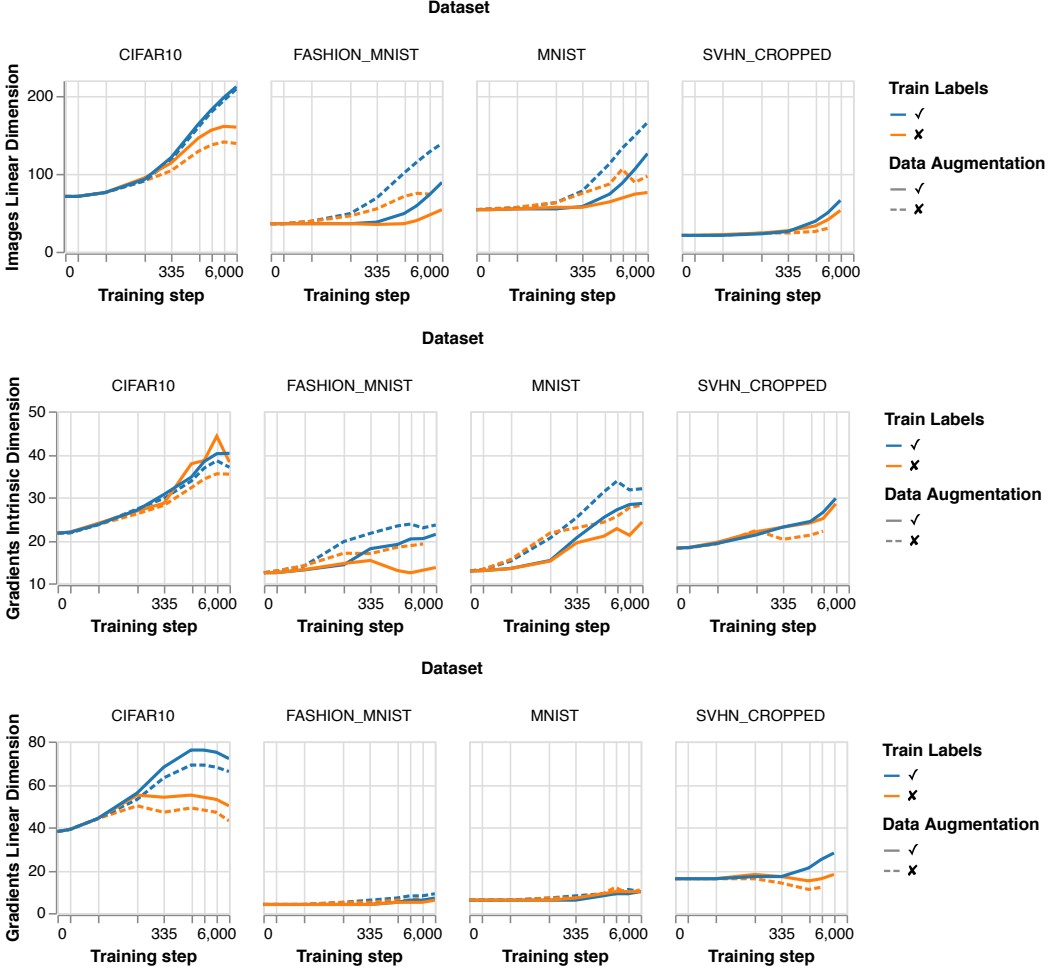

Figure A3: **Dimensionality of learned dataset manifolds grows with training.** Similarly to Figure 7, we show that dimensionality of the learned dataset grows with training as measured by **(top)** linear dimension, **(middle)** intrinsic dimension of infinite-width gradients (a.k.a. NTK features), and **(bottom)** linear dimension of gradients. Settings: 500 images, no ZCA.

Table A2: **Training on full CIFAR-10 train data with ConvNet neural network.** Regularized ZCA, presence of instance normalization layers, augmentation settings, and loss function (mean squared error and cross-entropy) are varied.

| ZCA | Normalization | Data Augmentation | Test Accuracy, % |
|:---:|:---:|:---:|:---|
| | | | MSE: 76.9±0.8
XENT: 80.4±0.2 |
| | ✓ | | MSE: 78.8±3.7
XENT: 84.9±0.6 |
| ✓ | | | MSE: 82.1±0.2
XENT: 84.4±0.1 |
| ✓ | ✓ | | MSE: 85.4±1.5
XENT: 86.3±0.9 |
| | | ✓ | MSE: 86.6±0.2
XENT: 88.9±0.2 |

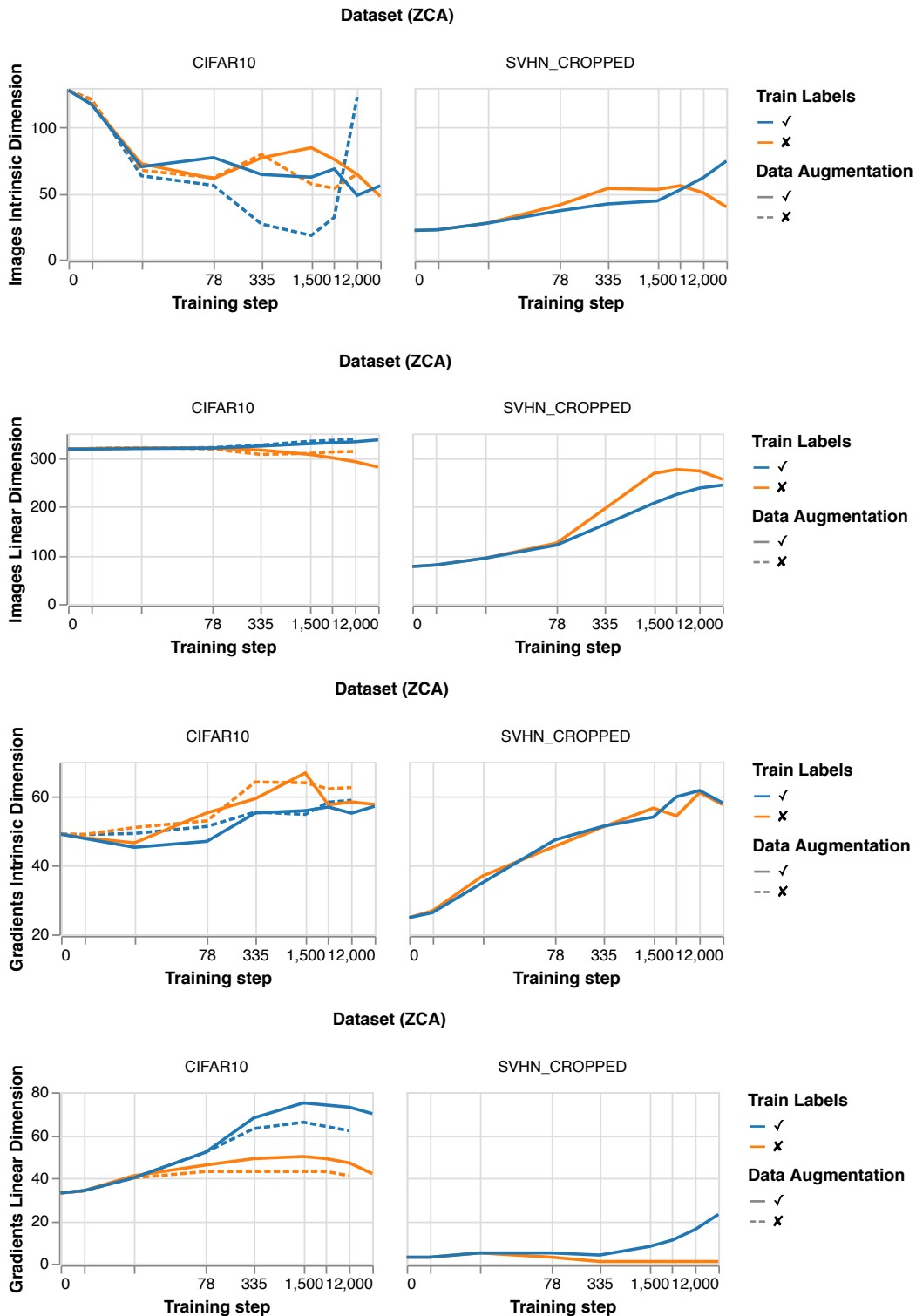

Figure A4: **Change in dimensionality of learned ZCA preprocessed dataset manifolds during training.** Similarly to Figure 7 and A3, dimensionality of the learned dataset generally grows with training as measured by (top to bottom) intrinsic dimension, linear dimension, intrinsic dimension of infinite-width gradients (a.k.a. NTK features), and linear dimension of gradients. However, we remark that the trend is less robust in the ZCA setting, notably the dataset at initialization can have a much higher dimension ($y$-value at step 0) than without ZCA-preprocessing, and can decrease during training (e.g. top-left plot). Settings: 500 images.

Table A3: **KIP Training for MNIST.** Complete set of training options for KIP and the corresponding transfer performance to neural network training (NN column). † denotes best chosen transfer is obtained with XENT loss rather than MSE.

| Imgs/Class | Train Labels | Data Augmentation | KIP Accuracy, % | NN Accuracy, % |
|---|---|---|---|---|
| 1 | ✓ | ✓ | 96.5±0.0 | 82.9±0.3 |
| | ✓ | | 97.3±0.0 | 82.9±0.4 |
| | | ✓ | 95.2±0.2 | 87.2±0.2[†] |
| | | | 95.1±0.1 | 90.1±0.1[†] |
| 10 | ✓ | ✓ | 99.1±0.0 | 95.5±0.0 |
| | ✓ | | 99.1±0.0 | 95.8±0.2 |
| | | ✓ | 98.4±0.0 | 96.6±0.0 |
| | | | 98.4±0.0 | 97.5±0.0 |
| 50 | ✓ | ✓ | 99.5±0.0 | 98.1±0.1 |
| | ✓ | | 99.4±0.0 | 98.2±0.2 |
| | | ✓ | 99.1±0.0 | 98.1±0.1 |
| | | | 99.0±0.0 | 98.3±0.1 |

Table A4: **KIP Training for Fashion-MNIST.** Complete set of training options for KIP and the corresponding transfer performance to neural network training (NN column).

| Imgs/Class | Train Labels | Data Augmentation | KIP Accuracy, % | NN Accuracy, % |
|---|---|---|---|---|
| 1 | ✓ | ✓ | 76.7±0.2 | 67.3±0.7 |
| | ✓ | | 82.9±0.2 | 73.5±0.5 |
| | | ✓ | 76.6±0.1 | 69.8±0.1 |
| | | | 78.9±0.2 | 72.1±0.3 |
| 10 | ✓ | ✓ | 88.8±0.1 | 81.9±0.1 |
| | ✓ | | 91.0±0.1 | 85.1±0.2 |
| | | ✓ | 85.3±0.1 | 82.5±0.2 |
| | | | 87.6±0.1 | 86.8±0.1 |
| 50 | ✓ | ✓ | 91.0±0.1 | 85.7±0.1 |
| | ✓ | | 92.4±0.1 | 88.0±0.1 |
| | | ✓ | 88.1±0.2 | 84.0±0.1 |
| | | | 90.0±0.1 | 86.9±0.1 |

Table A5: **KIP Training for CIFAR-10.** Complete set of training options for KIP (whether to use ZCA regularization, label training, or augmentations) and the corresponding transfer performance to neural network training (NN column).

| Imgs/Class | ZCA | Train Labels | Data Augmentation | KIP Accuracy, % | NN Accuracy, % |
|---|---|---|---|---|---|
| | ✓ | ✓ | ✓ | 63.4±0.1 | 48.7±0.3 |
| | ✓ | ✓ | | 64.7±0.2 | 47.9±0.1 |
| | ✓ | | ✓ | 58.1±0.2 | 49.5±0.4 |
| 1 | ✓ | | | 58.5±0.4 | 49.9±0.2 |
| | | ✓ | ✓ | 55.1±0.5 | 34.8±0.4 |
| | | ✓ | | 56.4±0.4 | 36.7±0.5 |
| | | | ✓ | 50.1±0.1 | 35.3±0.5 |
| | | | | 50.7±0.1 | 38.6±0.4 |
| | ✓ | ✓ | ✓ | 75.5±0.1 | 59.4±0.0 |
| | ✓ | ✓ | | 75.6±0.2 | 58.9±0.1 |
| | ✓ | | ✓ | 66.5±0.3 | 62.6±0.2 |
| 10 | ✓ | | | 67.6±0.3 | 62.7±0.3 |
| | | ✓ | ✓ | 69.3±0.3 | 45.6±0.1 |
| | | ✓ | | 69.6±0.2 | 47.4±0.1 |
| | | | ✓ | 60.4±0.2 | 47.7±0.1 |
| | | | | 61.0±0.2 | 49.2±0.1 |
| | ✓ | ✓ | ✓ | 80.6±0.1 | 64.9±0.2 |
| | ✓ | ✓ | | 78.4±0.3 | 66.1±0.1 |
| | ✓ | | ✓ | 71.4±0.1 | 67.7±0.1 |
| 50 | ✓ | | | 72.5±0.2 | 68.6±0.2 |
| | | ✓ | ✓ | 74.8±0.3 | 55.0±0.1 |
| | | ✓ | | 72.1±0.2 | 55.8±0.2 |
| | | | ✓ | 66.8±0.1 | 56.1±0.2 |
| | | | | 67.2±0.2 | 56.7±0.3 |

Table A6: **Label Solve on CIFAR-10.** We consider both ZCA and no ZCA preprocessing.

| Imgs/Class | ZCA | LS Accuracy, % | NN Accuracy, % |
|---|---|---|---|
| 1 | ✓ | 26.1 | 24.7±0.1 |
| | | 26.3 | 22.9±0.3 |
| 10 | ✓ | 53.6 | 49.3±0.1 |
| | | 46.3 | 41.5±0.2 |
| 50 | ✓ | 65.9 | 62.0±0.2 |
| | | 57.4 | 49.0±0.2 |

Table A7: **KIP Training for SVHN.** Complete set of training options for KIP (whether to use ZCA regularization, label training, or augmentations) and the corresponding transfer performance to neural network training (NN column). † denotes best chosen transfer is obtained with XENT loss rather than MSE.

| Imgs/Class | ZCA | Train Labels | Data Augmentation | KIP Accuracy, % | NN Accuracy, % |
|---|---|---|---|---|---|
| | ✓ | ✓ | ✓ | 64.3±0.4 | 57.3±0.1 |
| | ✓ | ✓ | | 62.4±0.2 | - |
| | ✓ | | ✓ | 48.0±1.1 | 44.5±0.2 |
| 1 | ✓ | | | 48.1±0.7 | - |
| | | ✓ | ✓ | 54.5±0.7 | 23.4± 0.3 |
| | | ✓ | | 52.6±1.1 | 39.5± 0.4 |
| | | | ✓ | 40.0±0.5 | 19.6±0.0 |
| | | | | 40.2±0.5 | 20.3±0.2$^{\dagger}$ |
| | ✓ | ✓ | ✓ | 81.1±0.6 | 74.2 ± 0.2 |
| | ✓ | ✓ | | 79.3±0.1 | - |
| | ✓ | | ✓ | 75.8±0.1 | 75.0± 0.1 |
| 10 | ✓ | | | 64.1±0.3 | - |
| | | ✓ | ✓ | 80.4±0.3 | 57.3±1.5 |
| | | ✓ | | 79.3±0.1 | 59.8±0.3 |
| | | | ✓ | 77.5±0.3 | 59.9± 0.3 |
| | | | | 76.5±0.3 | 64.2±0.3 |
| | ✓ | ✓ | ✓ | 84.3±0.2 | 78.4±0.5 |
| | ✓ | ✓ | | 82.0±0.1 | - |
| | ✓ | | ✓ | 81.3±0.2 | 80.5±0.1 |
| 50 | ✓ | | | 72.4±0.3 | - |
| | | ✓ | ✓ | 84.0±0.3 | 71.0±0.4 |
| | | ✓ | | 82.1±0.1 | 71.2±1.0 |
| | | | ✓ | 81.7±0.2 | 72.7±0.4$^{\dagger}$ |
| | | | | 80.8±0.2 | 73.2± 0.3$^{\dagger}$ |

Table A8: **Label Solve on SVHN.** We consider both ZCA and no ZCA preprocessing.

| Imgs/Class | ZCA | LS Accuracy, % | NN Accuracy, % |
|---|---|---|---|
| 1 | ✓ | 23.9 | 23.8±0.2 |
| | | 21.1 | 20.0±0.2 |
| 10 | ✓ | 52.8 | 53.2±0.2 |
| | | 40.1 | 37.9±0.2 |
| 50 | ✓ | 76.8 | 76.5±0.3 |
| | | 69.2 | 66.3±0.1 |

Table A9: **KIP Training for CIFAR-100.** Complete set of training options for KIP (whether to use ZCA regularization, label training, or augmentations) and the corresponding transfer performance to neural network training (NN column). † denotes best chosen transfer is obtained with XENT loss rather than MSE.

| Imgs/Class | ZCA | Train Labels | Data Augmentation | KIP Accuracy, % | NN Accuracy, % |
|---|---|---|---|---|---|
| | ✓ | ✓ | ✓ | 33.3±0.3 | 15.7±0.2 |
| | ✓ | ✓ | | 34.9±0.1 | - |
| | ✓ | | ✓ | 30.0±0.2 | 10.8±0.2 |
| 1 | ✓ | | | 31.8±0.3 | - |
| | | ✓ | ✓ | 26.2±0.1 | 13.4±0.4 |
| | | ✓ | | 28.5±0.1 | - |
| | | | ✓ | 18.8±0.2 | 8.6±0.1† |
| | | | | 22.0±0.3 | - |
| | ✓ | ✓ | ✓ | 51.2±0.2 | 24.2±0.1 |
| | ✓ | ✓ | | 47.9±0.2 | - |
| | ✓ | | ✓ | 45.2±0.2 | 28.3±0.1 |
| 10 | ✓ | | | 46.0±0.2 | - |
| | | ✓ | ✓ | 41.4±0.2 | 20.7±0.4 |
| | | ✓ | | 42.5±0.3 | - |
| | | | ✓ | 37.2±0.1 | 18.6±0.1† |
| | | | | 38.4±0.2 | - |

Table A10: **Label Solve on CIFAR-100.** We consider both ZCA and no ZCA preprocessing.

| Imgs/Class | ZCA | LS Accuracy, % | NN Accuracy, % |
|---|---|---|---|
| 1 | ✓ | 23.8 | 11.8±0.2 |
| | | 18.1 | 11.2±0.3 |
| 10 | ✓ | 39.2 | 25.0±0.1 |
| | | 31.3 | 20.0±0.1 |

Table A11: **KIP Training for CIFAR-10 on shallower ConvNet.** Complete set of training options for KIP (whether to use ZCA regularization, label training, or augmentations). Here, we remove the prepended convolutional and ReLu layer from ConvNet used in other tables.

| Imgs/Class | ZCA | Train Labels | Data Augmentation | KIP Accuracy, % |
|---|---|---|---|---|
| 1 | ✓ | ✓ | ✓ | $62.6\pm0.2$ |
|  | ✓ | ✓ |  | $65.8\pm0.2$ |
|  | ✓ |  | ✓ | $58.7\pm0.5$ |
|  | ✓ |  |  | $59.1\pm0.4$ |
|  |  | ✓ | ✓ | $53.4\pm0.3$ |
|  |  | ✓ |  | $56.3\pm0.3$ |
|  |  |  | ✓ | $48.7\pm0.6$ |
|  |  |  |  | $50.1\pm0.2$ |
| 10 | ✓ | ✓ | ✓ | $74.5\pm0.3$ |
|  | ✓ | ✓ |  | $74.4\pm0.2$ |
|  | ✓ |  | ✓ | $65.9\pm0.1$ |
|  | ✓ |  |  | $67.0\pm0.4$ |
|  |  | ✓ | ✓ | $68.3\pm0.1$ |
|  |  | ✓ |  | $68.7\pm0.2$ |
|  |  |  | ✓ | $59.6\pm0.2$ |
|  |  |  |  | $60.8\pm0.2$ |
| 50 | ✓ | ✓ | ✓ | $79.6\pm0.2$ |
|  | ✓ | ✓ |  | $76.5\pm0.1$ |
|  | ✓ |  | ✓ | $70.4\pm0.1$ |
|  | ✓ |  |  | $71.7\pm0.2$ |
|  |  | ✓ | ✓ | $73.7\pm0.1$ |
|  |  | ✓ |  | $71.5\pm0.3$ |
|  |  |  | ✓ | $65.9\pm0.2$ |
|  |  |  |  | $66.9\pm0.2$ |

Table A12: **Label Solve on CIFAR-10 on shallower ConvNet.** We consider both ZCA and no ZCA preprocessing. Here, we remove the prepended convolutional and ReLu layer from ConvNet used in other tables.

| Imgs/Class | ZCA | LS Accuracy, % |
|---|---|---|
| 1 | ✓ | 26.5 |
|  |  | 26.4 |
| 10 | ✓ | 52.9 |
|  |  | 46.4 |
| 50 | ✓ | 65.5 |
|  |  | 56.9 |