# OpenReview forum: "Dataset Distillation with Infinitely Wide Convolutional Networks"
_NeurIPS.cc/2021/Conference — NeurIPS 2021 Poster_

### Official Review · Reviewer_xee8 · 2021-07-15

**Rating:** 5
**Confidence:** 3

**Summary:**

The paper proposes to use infinitely wide CNNs to perform dataset distillation – obtaining a tiny dataset that enables high accuracy. The approach is an extension of a prior work (Kernel Inducing Points (KIP), etc.). Results are shown on MNIST, Fashion-MNIST, SVHN, CIFAR-10/100 datasets.

**Ethical Concerns:**

None.

**Limitations And Societal Impact:**

Yes.

**Main Review:**

The paper has following strengths:
1.	Results for ConvNet (3-layer ReLU CNNs) outperform prior models.
2.	Results for transferring to NN show good results for 1-sample distilled datasets.
3.	Preliminary but interesting study of generated images is presented.

The paper has several major weaknesses:
1.	It is unclear if this proposed method will lead to any improvement for hyper-parameter search or NAS kind of works for large scale datasets since even going from CIFAR-10 to CIFAR-100, the model's performance reduced below prior art (if #samples are beyond 1). Hence, it is unlikely that this will help tasks like NAS with ImageNet dataset.
2.	There is no actual new algorithmic or research contribution in this paper. The paper uses the methods of [Nguyen et al., 2021] directly. The only contribution seems to be running large-scale experiments of the same methods. However, compared to [Nguyen et al., 2021], it seems that there are some qualitative differences in the obtained images as well (lines 173-175). The authors do not clearly explain what these differences are, or why there are any differences at all (since the approach is identical). The only thing reviewer could understand is that this is due to ZCA preprocessing which does not sound like a major contribution.
3.	The approach section is missing in the main paper. The reviewer did go through the “parallelization descriptions” in the supplementary material but the supplementary should be used more like additional information and not as an extension to the paper as it is.

Timothy Nguyen, Zhourong Chen, and Jaehoon Lee. Dataset meta-learning from kernel ridge-regression. In International Conference on Learning Representations, 2021.



--------------------
Update: Please see my comment below. I have increased the score from 3 to 5.


**Time Spent Reviewing:**

4

---

> ### Author Response · Authors · 2021-08-10
> **Response to Reviewer xee8**
>
> We thank the reviewer for their time and feedback.
> * For the strengths #2, note that we also achieve strong (and often SOTA) neural network transfer for other numbers of img per class ratios (see bold numbers in Table 2).
> * We would like the reviewer to clarify weakness #1. We achieve SOTA neural network transfer in 1, 10, 50 images per class on CIFAR10 and 1 image per class on CIFAR100 (only 10 image per class did worse than prior art). This results in four out of five of the numbers being SOTA and corroborates the strength of our method. We also would like clarification of what absolute performance has to do with NAS - they are a priori independent considerations. For instance, a dataset distillation method could not be SOTA but be more strongly correlated with performance on the entire training data.
> * Regarding weakness #2 on novelty, we would like to emphasize that there are several novel aspects of our work:
>     1) While our main algorithm is directly from [1], the engineering novelty of our work is significant and cannot be overlooked. It was only through both the coordination of hundreds of GPUs and the use of `jax.vjp` to make the backward differentiation efficient that the KIP training could be done feasibly with ConvNet. This took months of engineering and was not a straightforward implementation of [1], in which a single machine is sufficient and communication between workers is not required.  (Note that [1] primarily used the FullyConnected kernel for KIP, which trains several thousand times faster. No pooling layers were considered). Such engineering effort is justified by the fact that our work often improved upon prior work by surprisingly large margins (up to 37% gain in absolute test acc; SVHN 1img/cls). A modern theme in machine learning is to explore the benefits that come with scaling up (and implicitly, how they compare to algorithmic developments at a fixed scale). We believe our margin of improvement well exceeds the bar for how much either scaling up or algorithmic innovation improves performance, and thus worth sharing with the NeurIPS audience.
>     2) We believe that the empirical discovery of such highly performant datasets - for CIFAR-10, 10 images (0.02% of training dataset size) yields 64% test acc, or 500 images (1% of training dataset size) can yields above 80% test acc - is in itself surprising and novel. It suggests there is much to learn about optimizing features and representations, especially in conjunction with preprocessing methods (given the effectiveness of ZCA regularization).
>     3) Our analysis of the KIP images reveals qualitative and quantitative insights into what distinguishes distilled data from naturally occurring data. In particular, our observation that intrinsic dimension increases and that performance concentrates around low eigenvalues of the kernel matrix (instead of high ones) are surprising and suggests that KIP images are more complex than natural images in ways that deserve further study. Note that such increase in complexity is a novel unexpected finding. Reference [1] did not perform such analyses.
> * Regarding weakness #3, we will move the SM discussion on the parallelization using many GPUS to the main body of the paper.
>
> We hope we have addressed the reviewer’s main concerns and we would appreciate it if the reviewer could consider revising their score to accept the paper. If there are any remaining deficiencies, please let us know.
>
> References:
>
> [1] Nguyen et al., Dataset Meta-learning from Kernel Ridge-Regression, ICLR 2021

---

> > ### Comment · Reviewer_xee8 · 2021-08-24
> > **Response to authors' request for clarification**
> >
> > Regarding weakness 1, looking at the Table 2 results, it appeared that even though the approach worked very well for all cases for CIFAR-10 (1, 10, 50 samples per class), for a slightly more complex dataset like CIFAR-100, the approach outperforms prior work only for 1 sample per class. The authors spin this as "it is state of the art in 4 out of 5 cases", but to me, it appeared that the approach may not work if you go for more complex datasets. Why wasn't the 50 images per class experiment done for CIFAR-100? If the approach indeed works, that experiment would be very helpful. However, since I am late in responding to authors' clarification request, it would be unfair of me to penalize them for lack of this experiment. This is why, I am increasing score from 3 to 5. However, it should be noted that I am not convinced about Table 2 results in the current state.
> >
> > The remaining part of weakness 1 has to do with one of the core motivations of this work: how useful would dataset distillation be to trim down search space in Neural Architecture Search. My initial thought was if the proposed approach does not work very well for more complex datasets (as a continuation of my concern regarding CIFAR-100 results), then the images generated using this method may not be useful for NAS (or at least may not be more useful than other dataset distillation methods that outperform the current method). Upon thinking further, I think I understand authors' point that the dataset distillation result does not directly have anything to do with NAS. However, while "Dataset distillation may not achieve SotA but it may still be useful for NAS", the other direction is also true: "If the prior work on dataset distillation outperforms current method, it may be a better indicator for NAS than the current method". This is why I wasn't sure if the proposed method really brings any improvement for downstream tasks. However, since my arguments are related to CIFAR-100 dataset results (and I am late), and downstream tasks like NAS are beyond the scope of this work, I have decided to not penalize authors for this.
> >
> > About my other main concern, lack of novelty from ML/DL algorithms point of view: although I agree that the paper presents a few interesting observations, I do not think they are sufficient (the new insights represent a small portion of the paper). The other engineering novelty with 100s of GPUs is indeed hard and fruitful work. However, I am not sure if Neurips is the best venue for that work (maybe try MLSys?). On the other hand, if the ACs and other reviews do not think this is a concern, I would be okay with accepting this paper. But, in view of my current thinking, I cannot increase the score beyond 5. Should the paper be accepted, I strongly recommend the authors to include the 50 image per class experiment for CIFAR-100.

---

> > > ### Author Response · Authors · 2021-08-31
> > > **Response**
> > >
> > > First, we thank the reviewer for the response and seeing more of our contribution to increase the score. We’d like to use this response to clear up remaining different points of view.
> > >
> > >
> > > > Why wasn't the 50 images per class experiment done for CIFAR-100?
> > >
> > > One major reason the 50 img/class experiment for CIFAR-100 was not performed is due to the lack of competitive baselines to compare with. For example, neither of DC (Zhao et al., 2021), DSA (Zhao & Bilen 2021), KIP-FC (Nguyen et. al 2021), original Dataset Distillation (Wang et al., 2018) do not report results on such settings. Given dataset distillation is trying to obtain a small dataset with high compression ratio, 50 imgs/cls for CIFAR-100 would mean distilling to 10% of the dataset, while other 10 class datasets lead to ~1% compression, which one could argue to be more interesting.
> > >
> > > > the approach may not work if you go for more complex datasets.
> > >
> > > We politely disagree with the reviewer’s sentiment that “the approach may not work if you go for more complex datasets”. For a very loose definition of “complex”, one would argue CIFAR-10 (where our NN transfer achieves SoTA for all benchmarked settings) dataset to be more complex than digits (MNIST/SVHN) or Fashion-MNIST dataset. Also we note that, in general “performance change” during transfer from KIP to NNs increases as you increase # of img/cls which suggests “more complex” case NN transfer would be even better. On the other hand, note higher # of img/cls does not necessarily become “more complex” since in dataset distillation, it leads to more parameters to distill into.
> > >
> > > Nevertheless, motivated by the reviewer, we started training the 50 img/class on CIFAR-100. While we are still early in the training with several hundred KIP training iterations, we find that we have achieved 53.0% (49.1% w/o label learning) test accuracy and are still improving. For transfer to finite neural networks on these earlier datasets, we already obtained 40.5% test accuracy which would improve with further KIP training. We are happy to report these numbers (with more iterations) in the camera ready. As emphasized above, without a competitive baseline, whatever numbers we obtain will be SoTA but we can assure the reviewer that there’s nothing problematic generalizing to the CIFAR-100 50 img/class case.
> > >
> > > > However, it should be noted that I am not convinced about Table 2 results in the current state.
> > >
> > > Moreover we’d like to emphasize that while transfer to NN in Table 2 is an interesting downstream task, the true distillation benchmark itself is reported in Table 1 which achieves SoTA for all the settings. While Deep Learning is definitely an important and interesting topic as of now, as a scientific community we should also be aware of other ML methods which can perform a given task (dataset distillation) better. Our paper shows that using infinite-width correspondence, one can construct a kernel method that achieves SoTA on the dataset distillation tasks, at the same time obtaining transferability to finite NNs (in contrast, note that no prior NN-based distillation papers have demonstrated good transferability to other ML methods). It turns out even the downstream transfer also achieves SOTA in many cases which is good and surprising but should not discredit the superior results on the kernel side.
> > >
> > > To further elaborate the above point, below we consider doing inference with DC/DSA images using our kernel method (therefore the NN -> KIP transfer*):
> > >
> > >
> > > |  # Img / Cls | DC / DSA  | DC / DSA to KRR | DC/DSA perf change | KIP     | KIP to NN | KIP perf change
> > > |----------|---------|-------|-----|-------|--------|----------|
> > > | 1 | 28.3$\pm$0.5 / 28.8$\pm$0.7 | 29.6 / 27.6 | 1.3 / -0.8  |64.7$\pm$0.2 | 49.9 $\pm$ 0.2 | -9.2
> > > | 10 | 44.9$\pm$0.5 / 52.1$\pm$0.5 | 35.1 / 37.4 | -9.8 / -14.7 | 75.6$\pm$0.2 | 62.7 $\pm$ 0.3 | -4.6
> > > | 50 | 53.9$\pm$0.5 / 60.6$\pm$0.5 | 36.2 / 47.8 | -17.7 / -12.8  | 78.2$\pm$0.2 | 68.6 $\pm$ 0.2 | -4.5
> > >
> > >
> > > As you can see, there is also a substantial hit to performance, in fact a larger one than in the KIP -> NN transfer in Table 2.Therefore we argue that transferring distilled datasets between different ML methods (NN <-> Kernels) is an interesting open problem, but not a weakness of our method specifically.  (Note that for the last 3 columns in the above table, the KIP column uses label learning, KIP to NN uses KIP data without label learning, and perf change measures difference between KIP to NN and the corresponding KIP training without label learning. Hence the last column is not the difference between two previous columns.)
> > >
> > > *Note that we could not implement InstanceNorm (LayerNorm over pixels) in the infinite-width limit that DC/DSA used, but we have selected the maximum test accuracy over [no normalization, LayerNorm over channels, and LayerNorm over channels and pixels] (these settings have a closed-form infinite-width limit). All these settings performed similarly, so we believe this detail did not influence the conclusion of this experiment.
> > >
> > > > However, I am not sure if Neurips is the best venue for that work (maybe try MLSys?).
> > >
> > > Regarding our engineering contribution, we gently point out that in the NeurIPS call for papers (https://neurips.cc/Conferences/2021/CallForPapers), there is an explicit mention of `Infrastructure (implementations)` topic. Moreover it also says that `Machine learning is a rapidly evolving field, and so we welcome interdisciplinary submissions that do not fit neatly into existing categories`. We believe our work does have interdisciplinary contributions encompassing General Machine Learning, Deep Learning, Theory and Infra which we believe within the interest of the NeurIPS audience.
> > >
> > > We hope these answers help clarify any remaining concerns the reviewer may have.  Thank you!

---

### Official Review · Reviewer_Vxyf · 2021-07-18

**Rating:** 6
**Confidence:** 3

**Summary:**

The authors proposed a dataset distillation method using Kernel Inducing Points (KIP) and Label Solve (LS). The proposed method was designed for infinitely wide convolutional networks, however, when training a finite-width neural network model with the compressed dataset, the performance drop is small or moderate.

**Limitations And Societal Impact:**

Yes

**Main Review:**

1. The proposed method is very neat, as it is basically optimising the support set (training data for the training kernel matrix) for a kernel ridge regression problem.

2. The performance is reasonably good, even it is applied to the finite-width neural networks.

3. The discussion for the choice of lambda appears to be missing.

4. Line 235 -- 236, ", distributed metalearning framework that leverages hundreds of GPUs per training run" is confusing, can author elaborate which part of the model needs hundreds of GPUs and why?

**Time Spent Reviewing:**

2

---

> ### Author Response · Authors · 2021-08-10
> **Response to Reviewer Vxyf**
>
> We thank the reviewer for appreciating our work.
> * For point 3, the choice of lambda was explained in the experimental details of supplementary material (SM), section A, line 401; it is 1e-6.
> * For point 4, this was discussed in the SM (section B.1), but we will move it to the main body in the next revision. To give a brief summary: the expense of convolutional kernels with pooling (which scales quadratically with both the batch size and number of pixels) limits the kernel batch size to 18 on a Tesla V100 GPU (i.e. only an 18x18 block matrix can be computed per device at a time). For support size equal to 100 images and target batch size = 5K, we have to compute 500K kernel elements per train step, which is roughly 1.5K 18x18 kernel blocks worth of compute. This amounts to 480s worth of compute per train step. Hence a single train step requires 100-400 GPUS working in parallel in order to reduce a train step to the order of a few seconds.

---

### Official Review · Reviewer_HQy7 · 2021-07-20

**Rating:** 5
**Confidence:** 3

**Summary:**

Paper mainly extends the dataset distillation algorithms KIP and LS (Nguyen et al. 2021) to infinitely wide neural networks. To that end, authors present a distributed framework that draws upon huge hardware resources and show improvement on the distillation performance. Paper also analyzes the synthesized data samples (images, labels) via multiple studies.

**Limitations And Societal Impact:**

I do not see any potential negative societal impact of this work.

**Main Review:**

- Main drawback of the paper is its limited technical contributions. Authors extend the existing distillation methods to (primarily) infinite width neural networks.
- Also, while the proposed framework obtains SOTA results on multiple datasets, the transfer results (to finite width NNs) suffer performance drop and result in non SOTA results in 50% of the cases. However, the results are still very strong.
- The distributed meta-learning framework (one of the claimed contributions) developed for scaling the distillation algorithms to complex neural networks has only been discussed in appendix.
- Despite the above issues, the paper presents an important result of adapting the successful distillation algorithms to crucial neural network configurations. Further, the experimental analysis and studies performed on the distilled samples add strong value to the paper.

**Time Spent Reviewing:**

6-7

---

> ### Author Response · Authors · 2021-08-10
> **Response to Reviewer HQy7**
>
> We thank the reviewer for their time and useful suggestions.
>
> * Regarding our technical contributions, there are several.
>     1) While our main algorithm is directly from [1], the engineering novelty of our work is significant and cannot be overlooked. It was only through both the coordination of hundreds of GPUs and the use of `jax.vjp` to make the backward differentiation efficient that the KIP training could be done feasibly with ConvNet. This took months of engineering and was not a straightforward implementation of [1], in which a single machine is sufficient and communication between workers is not required. Such engineering effort is justified by the fact that our work often improved upon prior work by surprisingly large margins (up to 37% gain in absolute test acc; SVHN 1img/cls). A modern theme in machine learning is to explore the benefits that come with scaling up (and implicitly, how they compare to algorithmic developments at a fixed scale). We believe our margin of improvement well exceeds the bar for how much either scaling up or algorithmic innovation improves performance, and thus worth sharing with the NeurIPS audience.
>     2) Our analysis of the KIP images reveals quantitative insights into what distinguishes distilled data from naturally occurring data. In particular, our observation that intrinsic dimension increases and that performance concentrates around low eigenvalues of the kernel matrix (instead of high ones) are surprising and suggests that KIP images are more complex than natural images in ways that deserve further study. Note that such increase in complexity is a novel unexpected finding.
> * Regarding drop in performance for neural networks: Note that understanding how kernel and neural networks compare is an ongoing and important line of research. That our kernel results are SOTA across all datasets (outperforming neural network baselines) is therefore significant.
> * We plan to expand and move our discussion of the distributed meta-learning framework to the main paper according to the reviewer’s suggestion.
>
> We hope we have addressed the reviewer’s concerns and would appreciate it if the reviewer could consider revising their score accordingly.
>
> References:
>
> [1] Nguyen et al., Dataset Meta-learning from Kernel Ridge-Regression, ICLR 2021

---

> > ### Comment · Reviewer_HQy7 · 2021-08-28
> > **Rating post rebuttal**
> >
> > I have read all the reviews and rebuttal. I appreciate the authors' efforts in clarifying the reviewers' comments. However I feel the main contribution (although valuable) is not primarily a technical innovation, rather it is an engineering novelty that is not given the center stage in the paper by the authors. Hence I feel the paper may not be accepted for publishing at NeurIPS in the current form and retain my earlier score.

---

### Official Review · Reviewer_ET9m · 2021-07-21

**Rating:** 6
**Confidence:** 3

**Summary:**

In this work the authors extend the Kernel Inducing Points (KIP) method for dataset distillation to a new set of kernel functions and achieve new state of the art results on dataset compression. Since the KIP algorithm is a key algorithmic mechanism in this work, I will summarize it: The KIP method optimizes what can be seen as a test residual of a target dataset fitted by a kernel regression model that is created using a support dataset. This means that given a target dataset distribution, one can differentiate through the kernel computations to evolve the support data points in the support dataset freely in the input space and optimize them. After optimization the support dataset (Which is usually taken to be much smaller than the target distribution) can be used as a distilled version of the dataset. This compressed dataset enables classification and training without training with extensively large datasets. In this work the authors use the KIP algorithm using a family of kernels corresponding to infinitely wide neural networks that were discovered in the realm of NTK. Such infinite NTK convolution kernels are known to provide state of the art results (when compared with other kernels) on classic classification tasks such as MNIST, CIFAR, etc. The biggest issue of such infinite-network kernels is that they require immense amounts of computation to construct the kernel. In this work, the authors construct such kernels at each training step (!) and then back-propagate through the kernels (!!) to optimize the datapoint. This impressive feat is made possible by considering 3 layer convolutional neural networks and working with a relatively small number of data-points (5K as opposed to the whole dataset). This is still highly computationally challenging and the authors create a careful implementation including an orchestration system working with 100s of GPUs for optimizing the support dataset. With this new kernel family the authors achieve state of the art results on dataset compression and analyze the compressed dataset using high dimensional statistics tools.

**Limitations And Societal Impact:**

Large computational costs and focus on smaller datasets.

**Main Review:**

Significance:

This work presents new state of the art results for dataset distillation in the cases of simple classification datasets. Dataset distillation is an important topic for neural networks as it can enable various knowledge transfer methods via a shorter training loop. In addition the authors demonstrate some results on actual neural network transfer, since this is an important contribution, can the authors clarify and add in the main text what neural network architectures were used as the networks trained with the distilled datasets? Dataset distillation is most useful when the distilled dataset can be used among many different architectures. Can the authors share about experiments of using the distilled dataset to train different neural network architectures?

Since the authors utilize infinite-width networks kernels, the required computation for the method becomes very large and makes it naively infeasible. One of the key contributions of this work is the engineering component of this work which is highly non-trivial and is interesting and significant by itself.

Quality and clarity:

Overall this submission is very high quality with broad literature review, thorough experimental setup and in-depth analysis of the properties of the distilled datasets. Further the attached supplement provides ample details on the experimental setup in the appendix in addition to the new client-server set-up used for the large experiments.
Overall this is an interesting and well delivered work on dataset distillation but at the same time I am a bit torn due to some weaknesses. Some of the weaknesses of the work are the apparent lack of novelty of the idea, and irreproducibility (stemmed from the large amount of computation required). Further given the very large computational requirements of this method on simple datasets such as MNIST, and CIFAR I am concerned about the applicability of this method to larger datasets of interest. At this time I am going to assign this a score of (6) but if the authors can clarify my questions with regards to the transferability of the distilled datasets to different neural architectures I am willing to improve my score.


**Time Spent Reviewing:**

4

---

> ### Author Response · Authors · 2021-08-10
> **Response to Reviewer ET9m**
>
> * _Significance “Architectures used”_: For transfer to finite neural networks, the main architecture we used for most of the study, including the ablation studies, is ConvNet. This is mainly motivated by best potential transfer using the infinite-width correspondence. However, we also tested network transfer to Conv-Vec3, Conv-Vec8 (in Figure 1) and ConvNet with varying normalization layers as well as Myrtle5 (in Figure 2).
>
> * _Regarding “transferability”_:  Some analysis on general architecture transfer is shown in Figure (1, 2). For transfer, while moving away from architecture used during training does get a performance hit compared to using the same architecture, we still consistently observe performance gain compared to natural images. Thus information distilled to smaller datasets does transfer to different architectures. In our experiments, we also observe that kernel sampling training is a good way to make distilled datasets more robust even to architectures not used during KIP training.
>
> * _Quality and clarity “lack of novelty”_:
>     1) While our main algorithm is directly from [1], the engineering novelty of our work is significant and cannot be overlooked. It was only through both the coordination of hundreds of GPUs and the use of `jax.vjp` to make the backward differentiation efficient that the KIP training could be done feasibly with ConvNet. This took months of engineering and was not a straightforward implementation of [1], in which a single machine is sufficient and communication between workers is not required. Such engineering effort is justified by the fact that our work often improved upon prior work by surprisingly large margins (up to 37% gain in absolute test acc; SVHN 1img/cls). A modern theme in machine learning is to explore the benefits that come with scaling up (and implicitly, how they compare to algorithmic developments at a fixed scale).
>     2) We believe our margin of improvement well exceeds the bar for how much either scaling up or algorithmic innovation improves performance, and thus worth sharing with the NeurIPS audience.
> We believe that the empirical discovery of such highly performant datasets - for CIFAR-10, 10 images (0.02% of training dataset size) yields 64% test acc, or 500 images (1% of training dataset size) yields above 80% test acc - is in itself surprising and novel. It suggests there is much to learn about optimizing features and representations, especially in conjunction with preprocessing methods (given the effectiveness of ZCA regularization).
>     3) Our analysis of the KIP images reveals quantitative insights into what distinguishes distilled data from naturally occurring data. In particular, our observation that intrinsic dimension increases and that performance concentrates around low eigenvalues of the kernel matrix (instead of high ones) are surprising and suggests that KIP images are more complex than natural images in ways that deserve further study. Note that such increase in complexity is a novel unexpected finding.
> * Quality and clarity “irreproducibility”, “extension to larger datasets of interest”: We agree that scaling up to larger and more realistic datasets beyond academic benchmark is an important future direction. Note that the fields of dataset distillation and infinite-width networks are still young and so far most prior work benchmarked at most on datasets we’ve considered. For KIP using ConvNet, currently there are limitations for naively scaling up to ImageNet due to high image resolution and sheer extent of dataset size. Solving those challenges would be an exciting future direction.
> \
> &nbsp;
> \
> &nbsp;
>     One interesting idea for scaling to larger datasets includes using faster kernel approximation and inference such as sketching [2]. Such an approach can speed up kernel computation by a factor of several hundred.
> \
> &nbsp;
> \
> &nbsp;
>     Another possibility is to divide and conquer to handle larger support sets. For instance running KIP/LS on disjoint partitions of the target dataset and then taking the union of the distilled dataset as the distilled version of the target dataset. This would ameliorate the costs of kernel-ridge regression arising from cubic scaling in the support set size.
>
>     To mitigate some of irreproducibility concerns due to high computational requirements, we are in the process of open-sourcing our distilled datasets for the community to further investigate without needing to produce datasets themselves.
>
> References:
>
> [1] Nguyen et al., Dataset Meta-learning from Kernel Ridge-Regression, ICLR 2021
>
> [2] Zandieh et al., Scaling Neural Tangent Kernel via Sketching and Random Features. arxiv:2106.07880

---

### Author Response · Authors · 2021-08-31
**General comment on open sourced dataset**

Dear all,

In our submission, we mentioned that one of our contributions would be that we would open source the datasets we constructed (using 1000s of GPU hours). We are happy to report that the data is now available (over 760 checkpoints, spanning 38 different sets of hyperparameters, hosted through GitHub / GCS) and ready to be shared with the NeurIPS audience once the paper is de-anonymized. We feel it is worth pointing out that this is a valuable contribution to the research community by enabling the study of dataset distillation on data that would otherwise be inaccessible to those without large computational resources.

Best,
Authors

---

### Decision · Program_Chairs · 2021-09-28

**Decision:**

Accept (Poster)

**Comment:**

This paper introduces a distributed kernel-based meta-learning framework for dataset distillation. The paper extends previous work Nguyen et al. [2021] to a large-scale training setting.

The paper has received mixed reviews. The reviewers appreciate the significant improvement of dataset distillation performance on MNIST and CIFAR. However, several reviewers are concerned about the technical novelty compared to [Nguyen et al., 2021],  the scalability to large-scale datasets and large batch sizes, and the performance of downstream tasks. The authors have provided a rebuttal and addressed some of the concerns. The AC considers the improvement of dataset distillation on MNIST and CIFAR is significant compared to prior works. Before this paper, it’s hard to imagine that one can get 64.3% performance on CIFAR with 1 image per class for any machine learning algorithm. The AC also appreciates the engineering efforts and recommends that the authors move the distributed meta-learning algorithm from the appendix to the main text. The AC understands that the current method cannot handle ImageNet-level datasets. But given the low performance (29%) of previous dataset distillation methods even on CIFAR-10, this paper has presented a meaningful intermediate step towards the goal of dataset distillation. How to make it more efficient could be left as future work. Therefore,  the AC recommended accepting the paper.




**Consistency Experiment:**

NeurIPS has a long history of experimentation. In 2014, NeurIPS ran an experiment in which 10% of submissions were reviewed by two independent committees to quantify the randomness in the review process. This year, we repeated a variant of this experiment to see how the quality of the review process has changed over time.  This paper was part of the experiment and was therefore assigned to two committees (consisting of reviewers, an Area Chair, and a Senior Area Chair) that reached independent decisions.  If both committees made the same recommendation, this recommendation was followed. If a single committee recommended acceptance, the paper was accepted (with the exception of a few cases in which the other committee identified what we considered a fatal flaw, e.g., an error in a key result).

This copy’s committee reached the following decision: **Accept (Poster)**

The other committee assigned to the paper recommended **Reject**.  You can find the other set of reviews, along with any follow up discussion with the authors here:
https://openreview.net/forum?id=dBE8OI8_ZOa